# Synchronized seasonal excretion of multiple coronaviruses coincides with high rates of coinfection in immature bats

Alison J. Peel [1,2,3,15], Manuel Ruiz-Aravena [1,4,5,15], Karan Kim [6,15], Braden Scherting [7,8], Caylee A. Falvo [4], Daniel E. Crowley[4], Vincent J. Munster [9], Edward J. Annand [2,10], Karren Plain[2], Devin N. Jones-Slobodian[11], Tamika J. Lunn [12,13], Adrienne S. Dale[14], Andrew Hoegh [1,7] ✉, John-Sebastian Eden [3,6] ✉ & Raina K. Plowright [4] ✉

Bats host a high diversity of coronaviruses, including betacoronaviruses that have caused outbreaks and pandemics in humans and other species. Here, we study the spatiotemporal dynamics of co-circulating coronaviruses in *Pteropus spp* bats (flying foxes) in eastern Australia over a three-year period across five roost sites ($n = 2537$ fecal samples). In total, we identify six betacoronavirus clades, all within the nobecovirus subgenus. Genome sequencing supports overall clade assignments, however, also demonstrates the important role recombination has played in both the long-term and contemporary evolution of these viruses. Using a statistical framework that integrates individual and population level data, we assess the variability in prevalence of viral clades over space and time. Coronavirus infections and co-infections are highest among juveniles and subadults, particularly around the time of weaning. The overlapping shedding dynamics across multiple clades suggest opportunities for recombination, especially in younger bats. Understanding the ecological and host-viral drivers of these seasonally dynamic infections, co-infections, and recombination events will inform future predictive frameworks for coronavirus emergence in humans and other animals.

Zoonotic pathogen spillover occurs when a susceptible human host is exposed to a pathogen from an animal reservoir host, often via a bridging animal host[1]. If human-to-human transmission follows, spillovers can lead to outbreaks, epidemics, and ultimately, endemic circulation within human populations[2]. Genomic studies have established that most human coronaviruses originated from viruses in bats[3–5], including two endemic human coronaviruses causing mild respiratory and gastrointestinal infections[6] and the three recently emerged coronaviruses that have triggered devastating outbreaks and pandemics (SARS-CoV, MERS-CoV, and SARS-CoV-2)[5,7–9]. Preventing initial spillover events can stop pandemics at their source, saving lives and resources[10–12]. However, our understanding of the factors that drove

the original spillover of these coronaviruses remains limited, as most were historical or not investigated. Critical gaps in knowledge persist regarding contemporary coronavirus spillovers, including the drivers of the temporal and spatial dynamics of coronavirus shedding from bats and how these factors contribute to viral evolution and spillover risk within diverse bat coronavirus communities[13,14].

Coronaviruses have been detected in nearly all bat species tested[13,15]. Recent field investigations of wild bat populations reveal that the timing and magnitude of viral shedding varies with season[16–19], roost type[16,18,20,21], age and reproductive status[19,22–24], bat species[16,19,22,25,26], and virus variants[22–24,27], supporting earlier studies[13]. However, most studies rely on cross-sectional sampling, are

conducted over less than a year, or lack critical host metadata[13,15], limiting the ability to systematically evaluate the drivers of intra- and inter-annual variation in viral shedding that have been documented in other bat-virus systems[17,28–31]. To enhance our understanding of the spatial and temporal factors influencing coronavirus shedding in bats, long-term sampling across multiple years and locations is necessary.

To achieve large scale viral surveillance in bat populations, many studies rely on pooling samples, either during collection from underneath roosting sites or during screening. While this improves efficiency, it can overestimate prevalence[32,33] and prevents collection of individual-level metadata required to understand drivers of viral dynamics. Additionally, combining prevalence data from multiple viruses or clades obscures important details (see refs. 22,23,27). These common approaches limit our ability to assess interactions between clade-specific viral prevalence, rates of co-infections, and viral recombination, all of which may contribute to emergence of new variants with zoonotic potential[34,35].

Here, we analyse coronavirus shedding in co-roosting black flying foxes (BFF; *Pteropus alecto gouldi*) and grey-headed flying foxes (GHFF; *Pteropus poliocephalus*) in eastern Australia over a three-year period across five roost sites. We identify six betacoronavirus clades, all within the nobecovirus subgenus, and use whole genome sequencing of representative viruses to demonstrate the role of recombination in their evolutionary history. Using a modeling framework that integrates data from both individual bats and pooled samples collected from underneath roosts, we demonstrate distinct spatiotemporal dynamics of each clade. We show that this integrated approach combines the efficiency of pooled sampling with the inferential capabilities of estimation from metadata from individuals, which results in more accurate and precise estimates of prevalence than either sampling approach alone. Finally, we show that coronavirus infections and co-infections are highest among juvenile and subadult bats, particularly around the time of weaning. These co-infections during synchronous shedding across multiple clades suggest opportunities for recombination, especially in younger bats. Understanding the ecological and host-viral drivers of these seasonally dynamic infections, co-infections, and recombination events will inform future predictive frameworks for coronavirus emergence in humans and other animals.

## Results

To explore coronavirus prevalence and diversity, we screened samples using a pan-CoV RT-PCR-based metabarcoding approach across five study sites in southeast Queensland and northern New South Wales, Australia, where both BFF and GHFFs are known to roost (Supplementary Fig. 1). Following our model-guided screening framework (Supplementary Fig. 2), fecal samples were collected for viral screening directly from individual bats (Supplementary Fig. 3, Supplementary Table 1) and through population-level sampling by collecting excreta on plastic sheets placed under the roosts (under-roost sampling) (Supplementary Fig. 4, Supplementary Table 2). Coronavirus RNA was

detected in 98 of 1137 samples collected from individual bats (8.6%) and 71 of 510 pools collected from under roosts (13.9% of pools, comprising a total of 1392 under-roost samples). Overall clade specific prevalences ranged from 0 to 4.2% in individuals and from 0.4 to 7.6% in pools (Table 1). All clades were detected in under-roost pools; four of these clades were also detected in samples from BFF individuals (2d.ii, 2d.iv, 2d.v, and 2d.vi) and one clade (2d.iii) was detected in samples from GHFF individuals. Clade 2d.i was not detected in samples from individual bats.

Host mitochondrial (cytochrome b) sequences were then amplified for all CoV positive under-roost pools ($n = 72$) and a majority subset of the remaining CoV-negative pools ($n = 216$). Of the 201 under-roost pools that were successfully amplified and sequenced, BFF was the dominant species identified (98.5%, $n = 198/201$), with a similar rate between CoV-positive and negative pools. Consistent with our individual captures, field observations, and sample selection approach, these results confirm that our study cohort largely comprised BFF. Detection of GHFF DNA was rare in the under-roost pools (5.5%, $n = 11/201$) and GHFF DNA was mostly co-detected with BFF DNA ($n = 8/11$ GHFF positive pools). The only two clade 2d.iii pools that were successfully sequenced (of four total) were included in the GHFF detections. With the individual level results, this further confirmed the strong virus-host association between clade 2d.iii and GHFF.

### Phylogenetic position of viral clades

Through the deep sequencing of all RT-PCR positive specimens, six distinct RdRp genetic clades of betacoronavirus were identified (Supplementary Fig. 6). All six betacoronaviruses were within the nobecovirus subgenus with clades 2d.i-v falling with a larger group of AMB130-related nobecoviruses that have been mostly detected in *Pteropus* spp, while 2d.vi is related to the HKU-9 viruses (Fig. 1A). Across the target RdRp region, diversity was low within each nobecovirus clade (mean nucleotide identity 99.5%, range: 97.9–100%; Supplementary Fig. 7), but variable across different clades (mean nucleotide identity 72.7–89.7%). The 2d.i and 2d.ii viruses were most similar (89.7%, range: 89.3–90.0%) while the 2d.vi viruses were the most divergent from other clades (2d.i–2d.v) (72.7%; range: 70.0–74.1%). Three of the clades (2d.i, 2d.ii & 2d.iii) appeared novel with no closely related viruses on public databases (Fig. 1A), while 2d.iv formed a clade with other recently described nobecovirus strains, including CP07 from a grey headed flying fox (GHFF) in Sydney[36], Australia. Other 2d.iv viruses included PREDICT68-like nobecoviruses that have been identified across Indonesia and Thailand in *Pteropus* spp, including in a different black flying fox subspecies (*P. alecto alecto*). Similarly, the 2d.v viruses formed a clade with nobecoviruses from black flying foxes in Indonesia including the PREDICT67-like virus INDSWBT-131. The clade 2d.vi was genetically distinct from the clade 2d.i–2d.v viruses and grouped with the HKU9-like nobecoviruses that mostly contain viruses detected in *Rousettinae* bats. Although the apparent host-structure in the nobecovirus phylogeny (Fig. 1A) suggest that clade 2d.vi viruses were potentially not derived from *Pteropus*

**Table 1 | Overview of positive detections and samples tested from black flying foxes (BFF; *P. alecto gouldi*), grey-headed flying foxes (GHFF; *P. poliocephalus*) and pooled under-roost samples**

| Clade | Related viruses | BFF | GHFF | Under-roost pools | Total under-roost samples |
|---|---|---|---|---|---|
| 2d.i | Novel clade | 0/1108 (0.0%) | 0/29 (0.0%) | 3/510 (0.6%) | 1392 |
| 2d.ii | Novel clade | 20/1108 (1.8%) | 0/29 (0.0%) | 15/510 (2.9%) | 1392 |
| 2d.iii | Novel clade | 0/1108 (0.0%) | 9/29 (31.0%) | 4/510 (0.8%) | 1392 |
| 2d.iv | CP07-like | 34/1108 (3.1%) | 0/29 (0.0%) | 22/510 (4.3%) | 1392 |
| 2d.v | PREDICT67-like | 48/1108 (4.3%) | 0/29 (0.0%) | 40/510 (7.8%) | 1392 |
| 2d.vi | HKU9-like | 1/1108 (0.1%) | 0/29 (0.0%) | 2/510 (0.4%) | 1392 |

Related viruses: identifies whether detections formed a clade with existing known strains or represented novel clades with no known closely related viruses.

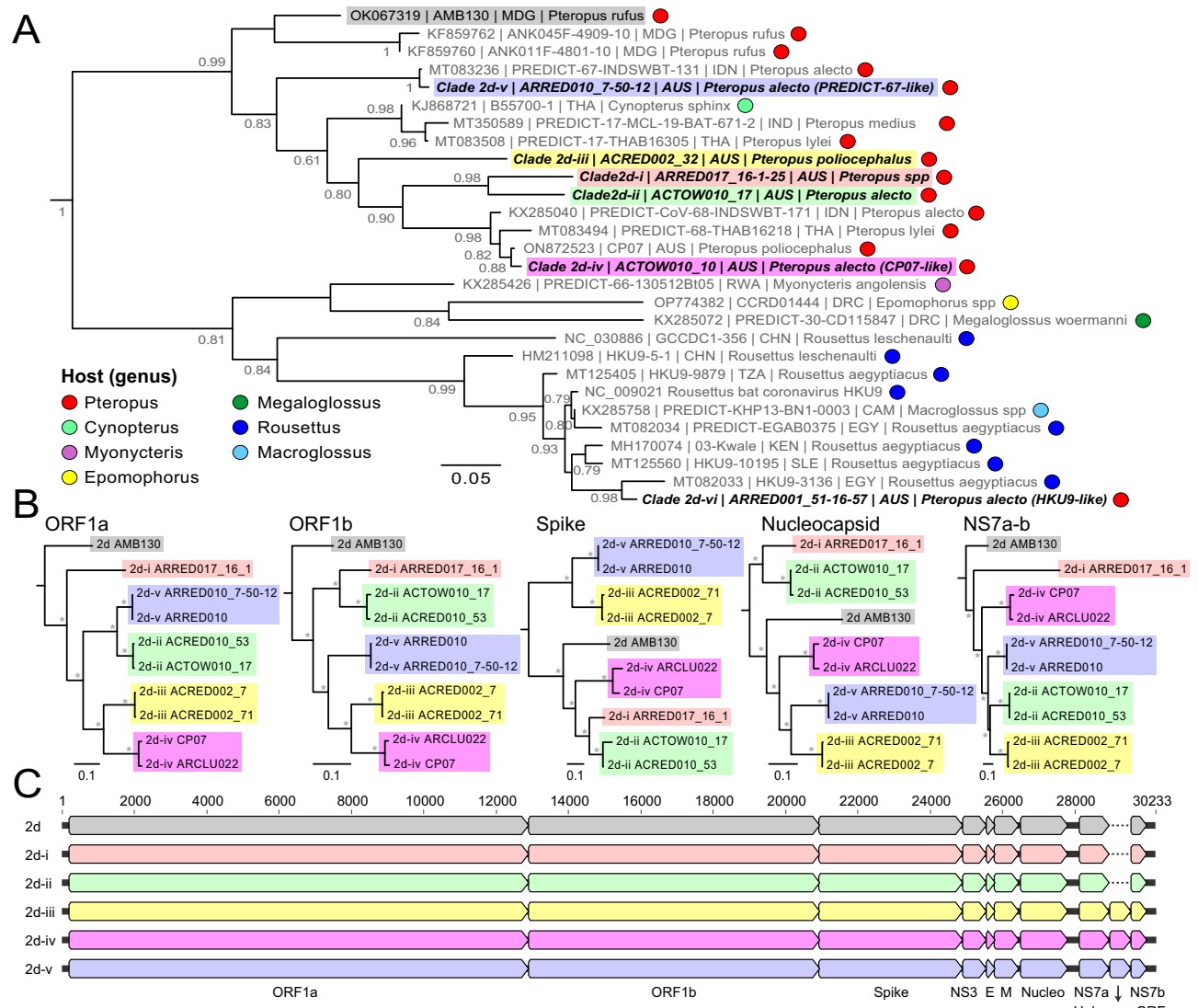

**Fig. 1 | Phylogenetic analysis of coronavirus (nobecovirus) clades identified in Australian flying foxes. A** Representative sequences from this study and NCBI GenBank were combined covering the target RdRp region (390nt - ORF1b) were codon aligned and analyzed using a maximum likelihood approach. Australian fruit bat clades described in this study are shown with bold and italicized text labels with colored boxes. Reference sequences are labelled as GenBank accession number, strain name, country three-letter code and host species. Colored circles indicate the host species (genus) as per the key provided. Branch support is indicated at node with SH-like values (>0.50) and all branches are scaled to the number of substitutions per site. **B** Major coding region phylogenies prepared from whole genome sequences of representative strains. The five phylogenies show the ORF1a, ORF1b, spike, nucleoprotein and NS7a-b coding regions with major clades colored as per (**A**) with the clade and strain name shown for each sequence. For all phylogenies, the topology was rooted using the Rousettus bat coronavirus HKU9. Branch support is provided at each node with a grey asterisk indicating SH-like values > 0.75 and all branches are scaled to the number of substitutions per site. **C** Genome alignment with coding regions of all 2d AMB130-like clades identified in this study. The scale shows the relative nucleotide position along the genome.

spp; the 2d.vi-positive sample in our study was collected directly from an individual BFF (*P. alecto gouldi*), confirming the host identity. A single deltacoronavirus was also detected (Supplementary Fig. 6), however was only detected in under-roost samples and subsequent RNA sequencing confirmed the most likely host as Australian white ibis (*Threskiornis molucca*) co-roosting with flying foxes, as *Pteropus* and *Threskiornis* mitochondrial sequences were co-detected in the sample (data not shown).

### Whole genome sequencing of representative strains

To further explore the evolutionary relationships of the betacoronaviruses identified in this study, we generated near-complete genomes from eight representative strains of the AMB130-like nobecoviruses (clades 2d.i–2d.v, Supplementary Table 3). Phylogenetic analysis of the major coding regions showed that the clades identified by the partial RdRp sequencing (Fig. 1A) were largely grouped across the different regions of the genome (Fig. 1B). To further refine the classification of the viruses identified here, we analysed conserved replication genes (3CLpro, NiRAN, RdRP, ZBD, and HEL1) following International Committee on Taxonomy of Viruses (ICTV) criteria, which suggested that clades 2d.i–2d.v were together members of a single species within the Nobecovirus subgenus, and distinct species from both AMB130 and HKU9. The basic genomic architecture of the virus clades was consistent except near the ends of the genome where some clades (2d.iii, 2d.iv and 2d.v) carry a possible additional ORF of unknown function between NS7a & NS7b (relative to AMB130) (Fig. 1C). The RdRp containing ORF1b appeared to be the most conserved (mean alignment nucleotide identity ± standard deviation: 84.4% ± 4.8%) with the spike and NS7a-b regions being most divergent (69.8% ± 10.1% & 55.3% ± 17.4%, respectively). In addition to the variable

divergence across the different coding regions, topological incongruence was also evident. For example, throughout the ORF1b, spike and nucleocapsid regions, 2d.i was most similar to the 2d.ii viruses, and generally appeared as a sister clade. However, in the ORF1a and NS7a-b regions, 2d.i was novel and distantly related to 2d.ii. We also saw changes in the relative branching order between the ORF1ab regions and the spike protein where the 2d.iii viruses shift between 2d.iv and

2d.v as closer related clades. This suggests a possible role for recombination in the diversification of the viruses, and while recombination analysis did propose possible recombination events near the junctions of the coding regions (ORF1a/b, ORF1b/spike and Nucleocapsid/NS7a; Supplementary Table 4), it is somewhat difficult to resolve if these are true recombination events or other evolutionary effects shaping the trajectory of the major clades. However, we did identify more well-supported circulating recombinant forms including a 2d.ii virus ACRED010_53 (from a juvenile BFF) that has most of its sequence being wild type 2d.ii, with a mosaic insertion of the spike N-terminal domain from a 2d.i virus (Supplementary Fig. 8), confirming recombination between the major nobecovirus clades. Together these data shows that the proposed RdRp clades represent well-established nobecovirus genetic lineages and that recombination is an important mechanism for viral diversification.

## Dynamics of circulation at the population level

We developed a Bayesian data integration approach to jointly estimate prevalence dynamics from pooled under-roost samples and individual catching data, while retaining information on individual bat-level characteristics relevant to transmission (Supplementary Fig. 9). In the integrated dataset, we detected coronavirus RNA in samples throughout the year (11/12 months), with higher prevalence between March and July across all 3 years of study primarily driven by overlapping excretion of viral clades 2d.iv and 2d.v (Figs. 2 and 3 and Supplementary Fig. 10). Both clades showed initial peaks in March (early-autumn), and detections of clade 2d.v extended into a second smaller peak in July (mid-winter). The estimated prevalence of the remaining clades was low, though clade 2d.ii demonstrated an early-autumn peak in prevalence in 2020 (synchronous with 2d.iv and 2d.v) and 2d.iii showed a sudden and substantial increase in winter 2018.

We used a model selection framework to evaluate support for competing hypotheses on clade- and location-specific prevalence dynamics. The most supported formulation was the clade-specific model (hypothesis 3, in methods), in which every viral clade exhibits a unique temporal dynamic yet has consistent patterns across sites in the study area. The difference in performance between the most supported model and the second-best model was more than 22 Information Criteria units, providing substantial evidence of better predictive capacity of the clade-specific model over models where prevalence dynamics were consistent across sites and clades, variable across both clades and sites, or consistent across clades but different among sites (Table 2).

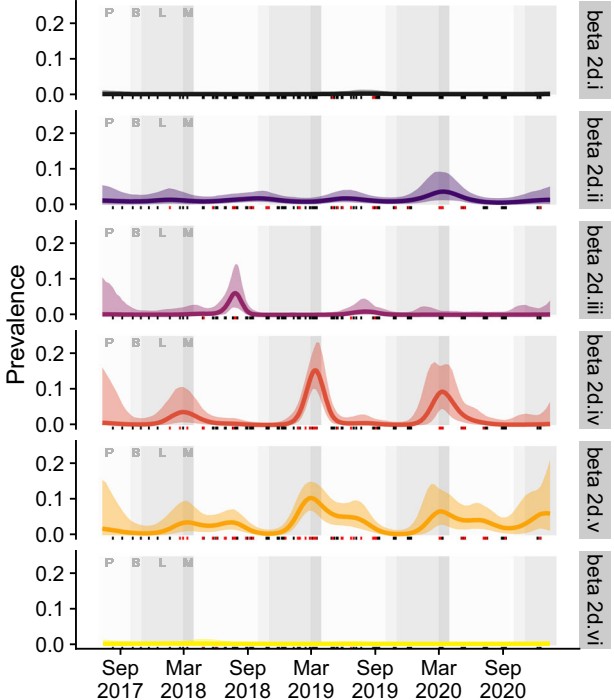

**Fig. 2 | Time-dependent shedding patterns at the population level for the six coronavirus clades identified, based on the most supported model using integration of both individual and population level information.** The background shading corresponds to annual cycles associated with bat behavior where P is pregnancy (April–September), B is births (October), L is lactation (November–March), and M is mating (March). The solid lines correspond to the posterior mean and the transparent bands are the 95% credible intervals. The tick marks across the bottom indicate periods of sampling where red marks are positive samples and black marks are negative.

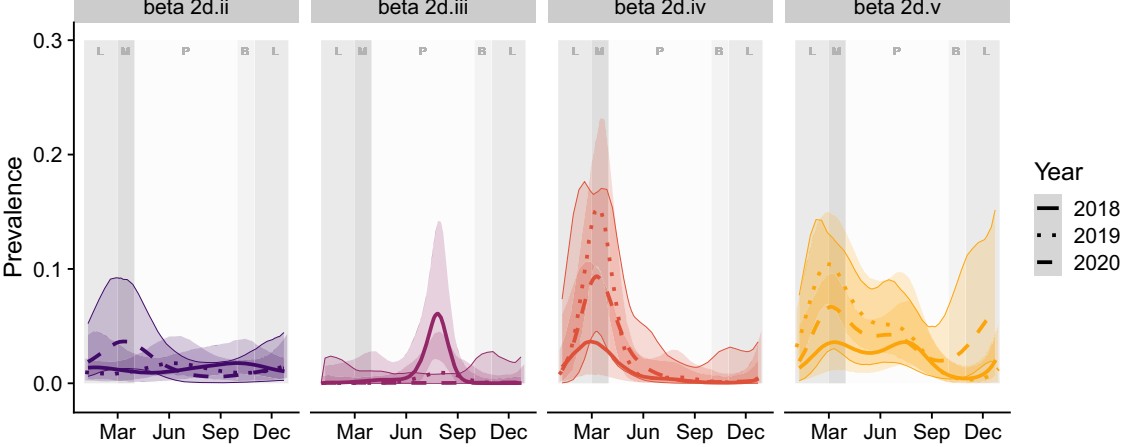

**Fig. 3 | Time-dependent shedding patterns at the population level for the four most identified coronavirus clades.** The background shading corresponds to annual cycles associated with bat behavior where P is pregnancy, B is births, L is lactation, and M is mating, as per Fig. 2. The lines correspond to the posterior mean for that year.

**Table 2 | The result of model selection, demonstrating strongest support for the clade-specific model, with delta Information Criteria (Δ IC) values more than 20 points less than the next best alternative**

| Hypothesis | Model | ELPD | LOOIC | ΔIC |
|---|---|---|---|---|
| 1 | All combined | −3591.4 | 1795.7 | +254 |
| 2 | Location-specific | −3592.0 | 1796.0 | +254.3 |
| 3 | Clade-specific | −3083.4 | 1541.7 | 0 |
| 4 | Clade-by-location | −3128.7 | 1564.6 | +22.9 |

Lower values of Leave-One-Out Information Criteria (LOOIC), or equivalently higher values of expected log pointwise predictive densities (ELPD) are preferred.

## Individual level dynamics of infection: dynamic binary regression

Using the individual-level dataset, we analyzed the relationship between individual bat information and viral detections to assess specific host conditions, namely age, sex, and species, associated with viral shedding. Positive coronavirus detections occurred in juvenile and subadults in all months (where $n > 1$). Our methodological prioritization for sampling BFF, the primary study species in Hendra virus surveillance, resulted in a significant disparity in sample sizes in our individual dataset (BFF, $n = 1108$; GHFF, $n = 29$ individuals). Within the BFF samples, detection rates were substantially higher in juveniles and subadults relative to adults (Fig. 4A and Supplementary Table 5). Sampling of GHFF individuals occurred within a restricted time frame (May–December 2018), and predominantly at the Redcliffe roost site ($n = 23$; Supplementary Fig. 3). We identified viral clade 2d.iii in 9/29 GHFF overall across all sites (9/19 GHFF sampled in the Redcliffe July 2018 session), but not in any of the 1108 individual BFF sampled (Fig. 4A, B). Conversely, we did not detect the other five viral clades in any individual GHFF. The lack of detection of the other viral clades in GHFF might be a consequence of the low sampling effort. However, considering the sampling effort of BFF, our results suggest the association of the viral clade of 2d.iii with GHFF. We note, however, the detection of viral clade 2d.iii in five under-roost sample pools (5/460 pools; two pools from the Toowoomba site, in one pool each in April 2018 and June 2019, and three pools from the Clunes site in August 2019). Even though field notes at the time of sampling indicated that GHFF were not present above the sheets represented in these positive pools, GHFF were observed co-roosting with BFF and may have contributed to the sheet prior to sample collection.

We used a dynamic binary regression model that accounts for temporal trends for each bat species and viral clade separately, determining the influence of age, sex, and species on shedding rates of the clades. Recognizing the pronounced differences in detections and sample sizes between the two bat species, we analyzed their shedding dynamics separately and limited modeling of GHFF results to the period when samples from this species were available. Using Leave-One-Out Information Criteria (LOOIC), a model with only age, rather than sex, or age and sex, was favored by nearly 60 Information Criteria units. This model showed with high probability that juveniles and subadults have higher prevalence than adults across clades 2d.ii, 2d.iv, and 2d.v, however significant differences in prevalence between juveniles and subadults were not supported (Fig. 5). The single positive result for clade 2d.vi meant that the estimated prevalence was nearly zero for the duration of sampling, and age-class differences and seasonality and could not be determined. By contrast, across the three clades with moderate (2d.ii) to high (2d.iv and 2d.v) prevalence, dynamics varied substantially across clades (Fig. 4C and Supplementary Fig. 11).

Tight autumn peaks in clade 2d.iv prevalence in juveniles and subadults corresponded with the first sampling event for juveniles

each year (~5 months of age; Supplementary Fig. 11). Similarly for clade 2d.v, the highest session-level prevalences were detected in juveniles in autumn (and in subadults in autumn–winter) and for both clades, these peak detections in juveniles and subadults aligned with peak detections in the under-roost dataset (Supplementary Figs. 10 and 11). However, while clade 2d.iv was rarely detected outside of seasonal peaks, clade 2d.v detections occurred throughout the year in adults, contributing to the overall flattening the clade 2d.v individual dynamic curves and distinct overall dynamics in individual infections between the two clades (Supplementary Fig. 11). Finally, clade 2d.ii dynamics presented mixed characteristics, with a more moderate overall prevalence and seasonality, with predominantly autumn–winter detections in juveniles and subadults, fewer aseasonal detections in adults, yet some summer seasonality in detections in under-roost samples (Supplementary Figs. 10 and 11).

## Recaptured and co-infected individuals

Nine of eleven individuals that were recaptured and had fecal samples available for screening across both captures were negative across both sampling events (Supplementary Table 6). The remaining individuals included one adult male BFF that tested negative to all clades in December 2019, then positive to clade 2d.ii in May 2020, and one subadult female BFF that tested positive to clade 2d.v in March 2020, then negative to all clades in July 2020.

Co-infection with multiple coronavirus clades were detected in 12 BFF ($n = 1108$, 1.1%), including one individual co-infected with three clades (Table 3). No co-infections were detected in GHFF ($n = 29$). Co-infections were primarily detected during the peak of viral circulation in autumn (10 out of 13, with the other three co-infections occurring soon before or after), and most (11/13) were detected in juvenile and subadult bats. Clades 2d.iv–2d.v were most frequently co-detected; indeed, in one March 2019 sampling session, more than half of all infected individuals had 2d.iv–2d.v co-infections (5/8). We examined the statistical associations between detections of the two most detected clades in individual BFF. Out of 1108 individuals, 34 individuals (3.1%) tested positive for 2d.iv, and 48 (4.3%) were positive for clade 2d.v; among these, ten individuals were co-infected with both clades (0.9% of all individuals; Table 4). The observed co-occurrence of both viral clades in samples from the same individuals was higher than expected given the overall prevalence of each clade ($\chi^2 = 53.23$, $p$ value = 0.0005). When also considering age, co-infection was especially pronounced in juveniles ($\chi^2 = 17.35$, $p = 0.001$) and subadult bats ($\chi^2 = 13.55$, $p = 0.008$), but not meaningful in adults ($\chi^2 = 1.29$, $p = 0.30$), suggesting a shared increased likelihood of infection associated with season and age-related factors (e.g., relative immune-naivety and immunocompetence)—and/or potential for biological interaction between them.

## Discussion

Bats are known to host a high diversity of coronaviruses, yet research integrating the dynamics, drivers, evolution, and interactions among co-circulating bat coronaviruses over time and space is limited. Our study addresses this gap by applying a statistical framework that integrates individual and population-level data to examine the spatiotemporal shedding of six distinct nobecovirus clades at five sites over three years. This approach, complemented by full viral genome sequencing, revealed distinct temporal dynamics for each coronavirus clade, characterized by periodic seasonal pulses of viral excretion that were consistent across locations. During these peak periods, juvenile and subadult bats had a higher prevalence of both single infections and co-infections compared to adults. We also found evidence of historical recombination across major clades and identified contemporary circulating recombinants. Our findings suggest that co-infection, and therefore recombination, is more likely among immature bats during periods of peak viral prevalence.

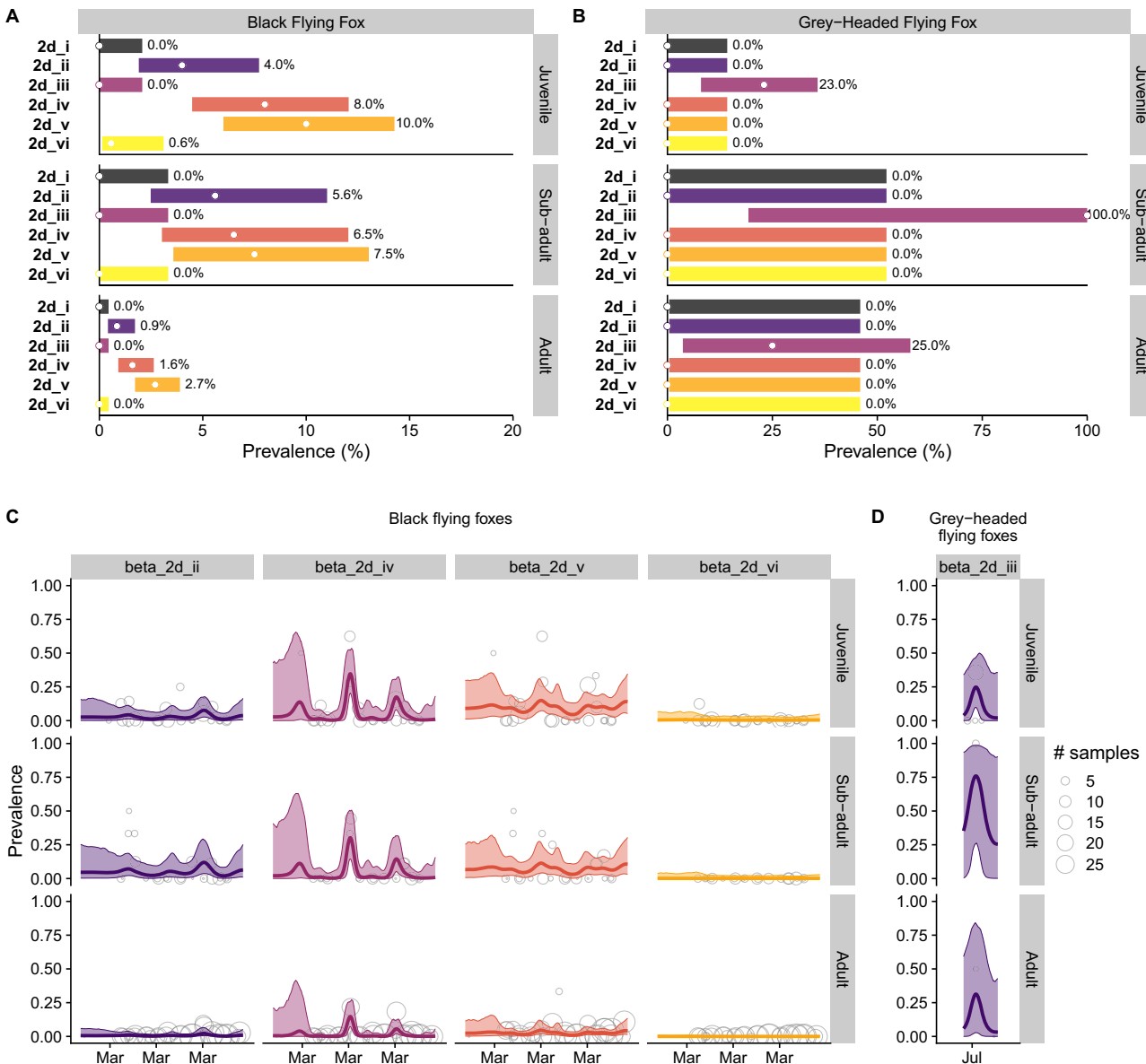

**Fig. 4 | Coronavirus prevalence and dynamics by host species and age class.**
**A**, **B** Empirical prevalence of the six coronavirus clades in individual bats, by species (BFF and GHFF) and age class is indicated with white dots and text. The uncertainty bars represent 95% Bayesian credible intervals using a uniform prior (Beta(1,1)) on prevalence. Note the different *x*-axis scale and large uncertainty levels, associated with small sample sizes for GHFF, particularly for adults and sub-adults, corresponding to the higher prevalence levels (sample sizes for BFF: 172 juvenile/ 105 subadult/822 adults; GHFF: 22/3/4). These figures are intended to present descriptive summaries of prevalence rather than for determining differences through statistical testing which would require formally controlling for testing multiple hypotheses. **C** Individual-level coronavirus prevalence dynamics (using individual data only). Estimated credible bands of prevalence from the dynamic

binary regression model for time-dependent shedding patterns for the four coronavirus clades identified in individual BFF by age category, inclusive of all sampling sites. No individual BFF were positive for clades 2d.i and 2d.iii throughout the entire sampling period and, thus, are omitted. The solid lines represent the posterior means, the shaded areas indicate credible intervals, the circles represent the raw prevalence within each sampling session, for each viral clade and specific age category combination. The circle size is proportionate to the sample size.
**D** Estimated credible bands of prevalence from the dynamic binary regression model for time-dependent shedding patterns for the coronavirus clade beta 2d.iii identified in individual GHFF by age category, inclusive of all sampling sites. Formatting as per (**C**).

Our genome analysis demonstrated that the six viral clades we detected could be clearly identified across the major coding regions, despite evidence of recombination. We detected varied evidence of species-specificity, and associations to existing known RdRp sequences that appeared to mimic some host phylogenetic relationships among *Pteropus* bats[37]. Of the clades we detected in BFF, three had previously been detected in Australian bats (2d.ii and 2d.v in BFF[38]; and 2d.iv in GHFF[36]). However, we also detected novel clades (2d.i in under-roost pools only, 2d.iii in GHFF and 2d.vi in BFF). The most frequently

detected clades in BFF (2d.iv and 2d.v) were most closely related to viruses previously detected in BFF subspecies *P. a. alecto* in Sulawesi, Indonesia[39]. Clade 2d.iii, unique to GHFF, appeared novel based on our partial RdRp and complete genome phylogenetic analyses; however, similar ORF1b and spike protein genomic fragments were reported from GHFF in Sydney, Australia (GenBank records ON872524-ON872526, with 97.9% pairwise identity between short read archive SRR19790909 and our ACRED002_71 2d.iii genome). This provides further evidence that 2d.iii may be a clade restricted to GHFF. While

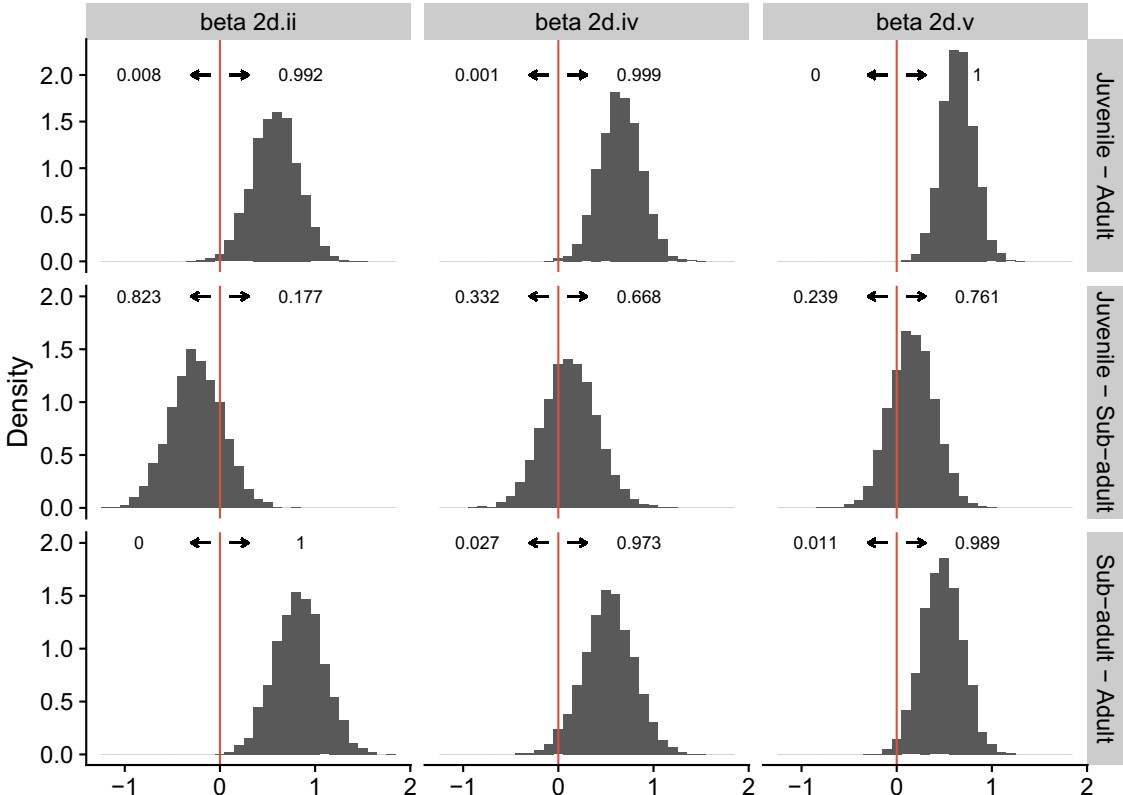

**Fig. 5 | Difference in coefficients for BFF, supporting higher prevalence in juveniles and subadults than adults.** The histograms represent posterior distributions for a contrast in the differences in overall prevalence between age groups for a clade. For example, the top left component of the figure suggests that overall prevalence of beta 2d.ii is larger for juveniles than adults with probability 0.992.

**Table 3 | Details of 13 black flying foxes with co-infections of clades 2d.ii, 2d.iv and 2d.v viruses, including age, sex, and site and date of sampling (dd/mm/yyyy)**

| Site | Age | Sex | Date | 2d.ii | 2d.iv | 2d.v |
|---|---|---|---|---|---|---|
| Toowoomba | Subadult | Male | 06/05/2020 | + | + | + |
| Toowoomba | Juvenile | Male | 16/05/2019 | + | + | |
| Redcliffe | Juvenile | Female | 28/07/2018 | + | | + |
| Redcliffe | Adult | Female | 05/03/2020 | + | | + |
| Clunes | Juvenile | Male | 20/02/2018 | | + | + |
| Redcliffe | Juvenile | Male | 09/03/2019 | | + | + |
| Redcliffe | Juvenile | Male | 09/03/2019 | | + | + |
| Redcliffe | Subadult | Male | 09/03/2019 | | + | + |
| Redcliffe | Juvenile | Female | 11/03/2019 | | + | + |
| Redcliffe | Juvenile | Female | 11/03/2019 | | + | + |
| Redcliffe | Juvenile | Female | 03/03/2020 | | + | + |
| Redcliffe | Subadult | Female | 14/05/2020 | | + | + |
| Redcliffe | Adult | Female | 08/07/2020 | | + | + |

The positive results are now shown with +.

**Table 4 | Observed counts of single detections of viral clade 2d.iv or viral clade 2d.v versus co-infection with both clades across age classes**

| | | 2d.v (−) | 2d.v (+) |
|---|---|---|---|
| Adult ($\chi^2 = 1.29$, $p = 0.26$) | 2d.iv (−) | 791 (794.3) | 21 (21.7) |
| | 2d.iv (+) | 12 (12.7) | 1 (0.3) |
| Subadult ($\chi^2 = 13.55$, $p = 0.0002$) | 2d.iv (−) | 95 (95.5) | 5 (7.5) |
| | 2d.iv (+) | 4 (6.5) | 3 (0.5) |
| Juvenile ($\chi^2 = 11.62$, $p = 0.0007$) | 2d.iv (−) | 148 (165.3) | 12 (16.7) |
| | 2d.iv (+) | 8 (12.7) | 8 (1.3) |

Expected counts assuming no association are shown in parentheses. These results, and the accompanying Chi-squared statistics, show elevated levels of co-infection in juvenile and subadult bats.

our results suggest potential viral-host-species co-evolution and ancestral host-viral relationships, limited full genome-scale data outside Australian *Pteropus* populations and with sufficient sampling in all species restricts detailed investigations at this stage.

Our models, integrating individual and pooled data while accounting for clade-specific spatiotemporal effects, demonstrated that time of year had a greater influence on shedding prevalence than location. Although shedding dynamics varied somewhat among clades, the consistent overlapping seasonality across clades suggests that shared, regular processes influence transmission dynamics. Specifically, the autumn-winter pulses of clades 2d.ii, 2d.iv, and 2d.v were predominantly driven by immature bats, peaking shortly after weaning, when juveniles are approximately 5 months old. The weaning period likely corresponds with the loss of maternal immunity and therefore introduction of susceptible individuals into the population[40]. Low-level detections continued in this cohort throughout winter and into their second year (as subadults). Subadults also exhibited higher prevalence during these pulses, with ongoing aseasonal detections potentially representing ongoing exposure of the naive cohort or increased susceptibility of animals with still-immature immune systems[41]. These observations are consistent with previous studies showing higher coronavirus shedding rates in younger animals and warrant experimental infection studies to better understand the role of maternal immunity and immune maturation to coronavirus infections in bats[19–21,24,30,41–44].

Our analyses suggest that clade-specific dynamics can vary and collapsing them into a single dataset may obscure important nuanced differences. For instance, clades 2d.iv and 2d.iii exhibited distinct seasonal peaks with most detections occurring within a one-month window (2d.iv: 40/54, 74%, February–March; 2d.iii: 13/14, 93%, July–August). By contrast, clade 2d.ii seasonality was less distinct, and clade 2d.v had an extended autumn–winter pulse with year-round detections. Although broader seasonal and age-related processes likely influence shared features of transmission dynamics among clades, these nuanced differences suggest unique host–viral interactions driving clade-specific transmission patterns[45]. The predominantly autumn peaks observed here also contrasts with the winter seasonality typically seen in Hendra virus and other paramyxoviruses within these same populations[46–50], consistent with the asynchronous shedding of paramyxoviruses and coronaviruses observed in other systems[51,52]. Larger sample sizes and in vitro studies are needed to interrogate the fine-scale differences in transmission dynamics and to identify specific factors influencing viral family- and clade-specific patterns detected here and in other systems[23,27].

Clade specific dynamics may help answer unresolved questions regarding how coronaviruses persist in bats populations between infection pulses, in the absence of discernible out-of-season shedding (e.g. 2d.iv dynamics here, and see also ref. [52]). In *Pteropus* species, the single annual birth pulse creates challenges for pathogen persistence because all susceptible individuals are introduced nearly simultaneously, potentially fuelling a rapid outbreak that exhausts remaining susceptibles[53]. Factors such as long infectious periods or sustained high population immunity[54–56] may lead to viral persistence at low prevalence. Our data, showing consistent low-level year-round detections in adults of the most prevalent clade (2d.v), supports the idea that some level of transmission persists between major shedding pulses and emphasises the importance of sampling throughout the year.

None of our recaptured bats were coronavirus-positive across multiple sampling sessions, precluding estimation of infectious periods. However, similar studies in cave-dwelling *Rousettus aegyptiacus* found infectious periods over a month in a very small proportion of recaptured bats[23] and longer infection durations in some individual bats contributed to the persistence of coronaviruses in *Myotis* spp[57] and Rhinolophus spp[51] populations.

The high rate of co-infections with clade 2d.iv and 2d.v, especially among younger bats, could result from synchronous high prevalence of both clades, or a facultative interaction between the clades that drives higher prevalence of each[34]. The occurrence of synchronous detections of viruses most closely related to 2d.iv and 2d.v (PREDICT CoV 67 and 68) in the Indonesian BFF subspecies (*P. a. alecto*)[39] supports this hypothesis of facultative interaction, however data is limited. Future research should focus on modeling the dynamics of these viral communities within populations and within-hosts, including facilitative to antagonistic immune responses during co-infections, and the implications of co-infections for disease outcomes and viral evolution through recombination.

Recombination is a key mechanism generating genetic diversity in coronaviruses, which can, in turn, enhance their potential for cross-species transmission[35,58–60]. For a recombination event to occur, two coronaviruses must co-infect the same cell in the same host at the same point of replication within the cell[35]. Our data confirm that co-infections predominantly occur during peak viral circulation of major clades in autumn, and within the demographic group with the highest prevalence of infection (immature bats), suggesting that these conditions represent the highest likelihood for recombination events. Within the major nobecoviruses clades identified here, the apparent topological incongruence across the genome-scale phylogenies suggested inter-clade recombination was likely an important force shaping viral diversification throughout their deep evolutionary history.

More importantly, our relatively small sample of eight representative genomes, identified at least one modern circulating recombinant form (here, a 2d.ii/2d.i recombinant virus), suggesting recombination is ongoing amongst these viruses and further genome sequencing will likely uncover more. Together, these results support the notion that high prevalence, and correspondingly, high rates of co-infection, increase the likelihood of recombination.

While spatial sampling remains crucial for capturing potential clade differences across broad geographic regions and habitat types[18,61] at the scale of our study area (>11,000 km²), we found that temporal surveillance rather than spatial surveillance was more informative for understanding coronavirus dynamics. This finding may be explained, in part, by the extreme mobility of *Pteropus* bats[62], and is therefore likely generalizable to other bat species known to travel long distances (such as annually migrating insectivorous bats)[63]. Temporal surveillance will also maximize the opportunity to detect all circulating clades, and to identify peak infection periods and recombinant viruses, enhancing our capacity for early warnings and One Health initiatives. Yet despite a large sample size and regular individual catching sessions, we were underpowered to examine detailed drivers of transmission within each clade, such the role of demographics and reproductive cycles in driving transmission.

This study had several limitations. First, we were generally unable to catch and sample juveniles prior to weaning because our sampling was timed to avoid stress during the birthing and early lactation phases (October–November) and pups are typically crèched (grouped together overnight) in roosts from late lactation until weaning. It is possible that peaks in viral shedding in this age group occurred before our first sampling in March. Although under-roost data did not indicate earlier peaks for 2d.iv and 2d.v, the detections of 2d.ii during spring and summer suggest that increased sampling of juveniles during the lactation period could reveal additional age-related shedding. While the use of a pan-CoV RT-PCR enables the detection of a wide diversity of strains, these approaches can lack some sensitivity compared to less degenerate primer sets, and consequently, our estimates of prevalence and re-infection rates might be reduced by false negatives. Furthermore, our approach lacked measurements of viral load. One study found higher loads of coronavirus in immature *Hipposideros* bats in Ghana[22], and Lunn et al.[31] reported that high viral load was associated with Hendra virus spillover in our study populations. Finally, we lacked tools for assessing immunity to coronaviruses in bats and serological assays for the specific circulating coronaviruses have not yet been developed, limiting our ability to test for evidence of waning immunity and re-infection.

The emergence of zoonotic coronaviruses in humans over recent decades highlights the need for preparedness against future outbreaks and pandemics. While various studies have identified environmental factors driving Hendra virus infection dynamics and spillover[28,49,50,55,64], our research extends this work to investigate the drivers of seasonally dynamic coronavirus infections, co-infections, and viral recombination. We show that juvenile and subadult bats play a key role in driving pulses of viral shedding and pose the highest risk for co-infections during these periods. Although the zoonotic potential of nobecoviruses remains uncertain, our findings suggest that the synchronized shedding of multiple coronavirus clades in younger bats represents a high-risk scenario for viral recombination and the emergence of novel viruses. This research advances our understanding of the ecological and host-related factors shaping coronavirus dynamics and evolution and identifies high-risk periods for recombination and shedding of coronaviruses in bat populations. Our work informs predictive frameworks for future zoonotic threats.

## Methods

Bat capture and sampling was performed following best practices[65]. Field protocols were approved by the Montana State University

Institutional Animal Care and Use Committee (201750) and Griffith University Animal Ethics Committee (ENV/10/16/AEC and ENV/07/20/AEC). The research was conducted in accordance with Scientific Purposes Permits from the Queensland Department of Environment and Heritage Protection (WISP17455716, WA0012532 and WA0058827), a permit to Take, Use, Keep or Interfere with Cultural or Natural Resources (Scientific Purpose) from the Department of National Parks, Sport and Racing (WITK18590417), and a Scientific License from the New South Wales Parks and Wildlife Service (SL101800).

## Study sites
We selected five study sites in south-east Queensland and northern New South Wales, Australia, where black flying foxes (BFF; *Pteropus alecto gouldi*) and grey-headed flying foxes (GHFF; *Pteropus poliocephalus*) were regularly present (Supplementary Fig. 1; Toowoomba (−27.60 S, 151.94 E), Redcliffe (−27.23 S, 153.10 E), Sunnybank (−27.58 S, 153.05 E) and Burleigh Knoll (−28.08 S, 153.44 E), in Queensland, Australia and Clunes (−28.73 S, 153.42 E), New South Wales (NSW)). Site selection criteria included attributes associated with viral spillover risk and feasibility[31], including: continuous occupation by BFF, recently overwintering, limited native winter food, and sampling feasibility, access, and permissions[31]. Little red flying-foxes (LRFF; *Pteropus scapulatus*) were seasonal visitors to the Redcliffe, Sunnybank, and Toowoomba sites (October–May). Another species that regularly roosted within the Redcliffe site was the Australian white ibis (*Threskiornis molucca*). We collected fecal samples for viral screening directly from individual bats (n = 1188) as well as from population-level sampling by collecting excreta on plastic sheets under the roosts (n = 1349).

## Individual bat sampling
Between May 2018 and December 2020, we caught and sampled flying foxes over 13 sessions at the Redcliffe roost and 15 sessions at the Toowoomba roost. Additionally, we caught and sampled at the Clunes roost in August 2017, February 2018, and August 2018 (Supplementary Table 1 and Supplementary Figs. 1 and 3). All 31 individual sampling sessions extended 3–4 consecutive days, with the objective of collecting samples from approximately 60 bats per session. Sampling sessions typically occurred five times per year and covered various stages of the reproductive cycle: in March (early-autumn, around weaning/mating), May (late-autumn, early-pregnancy), July (mid-winter, mid-pregnancy), September (early spring, late-pregnancy), and in December (early-summer, lactation), following a mid-October peak in births.

Bats were caught in mist nets (Ecotone 716/7-12P) set at or above roosting height with modified antenna poles (Spiderbeam 26 m Fiberglass Telescoping Pole) and opened approximately 2 h before dawn to catch individuals returning from their nocturnal foraging activities. Bats were removed from the nets immediately to minimize stress or injury. After capture, bats were temporarily placed in cloth bags until they were anesthetized with isoflurane (starting at 5% then reducing to 1.5%) for sampling (within 6 h). While our sampling primarily focused on BFF, a smaller number of GHFF were sampled opportunistically. All bats were checked for a Passive Integrated Transponder (PIT) tag to identify recaptured individuals. If no PIT tag was present, we inserted a PIT tag (RFID, ZD Tech Group China) under the skin between the scapulae while the bat was anesthetized.

The data recorded for each bat included species, sex, age category (juvenile, subadult, or adult), body mass (grams), forearm length (millimetres), and reproductive status (males: immature, reproductive; females: immature, pregnant, lactating, post-lactation). Age categories were assigned using a combination of sexually dimorphic ranges (weight and forearm, where applicable) in conjunction with the timing of seasonal birth pulses (juveniles: smaller than adult body size (generally <150 mm forearm and <450 g mass), not sexually mature, assumed to be <12 months of age (including dependent pups); subadults near or at adult body size, not sexually mature, assumed to be

12–24 months; adults: full body size (generally >160 mm forearm and >550 g mass), with females showing evidence of prior lactation (elongated nipples) and males with enlarged penis and descended, enlarged testes, assumed to be >24 months). Reproductive status was determined by evidence of sexual maturity (see above), and current or past reproduction. Specifically, pregnancy was determined by the presence of a palpable uterine bulge; lactation by the ability to express milk from the nipples, the size and shape of nipples, and the absence of fur around nipples (lactation: elongated bare nipples; post-lactation: elongated furred nipples; immature: small furred nipples); and reproduction in males by the size and descension of genitalia. For additional details on age class and reproductive status definitions and methods, see Pietromonaco et al.[66]. We collected fecal samples using sterile cotton swabs either as the bat defecated under anesthetic, directly from the rectum, or from feces deposited within the cloth bags during holding periods. These swabs were immediately placed in individual resealable plastic bags, placed on ice, and subsequently preserved at temperature of −20 °C or −80 °C until the laboratory processing and analysis were performed. Dependent (suckling) juveniles generally remain in roosts overnight during December and January, and so were not captured and sampled during these months (with one exception).

All bats were administered fluids (one part 2.5% glucose and 0.45% NaCl solution and one part Hartmann's solution) subcutaneously and monitored for at least one hour before release back into their roost. All animal handling was conducted under approval from the Griffith University Animal Ethics Committee (Certificate: ENV/10/16/AEC and ENV/07/20/AEC). Personal Protective Equipment and disinfection protocols followed best practice guidelines (e.g. IUCN Bat Specialist Group, 2021; Wildlife Health Australia, 2020). The field team collected individual bat data onto optical mark recognition (OMR) forms, which were processed using Scantron software, which exported data into CSV files for analyses.

## Under-roost sampling
In conjunction with individual bat sampling, we collected fecal samples monthly from underneath roosting bats from all five sites (Toowoomba, Redcliffe, Sunnybank, Burleigh Knoll and Clunes; Supplementary Table 2 and Supplementary Figs. 1 and 4). Under-roost sampling methods are described in full in Lunn et. al[31]. We deployed plastic sheets measuring 0.9 × 1.3 meters beneath the trees where bats were roosting[31,67]. We positioned the plastic sheets under the trees before the bats returned to the site at dawn. To avoid sampling the same individuals twice, we maintained a minimum distance of at least one meter between sheets preferentially targeting areas of the roost occupied by BFF. In instances where the roost size was too small for sheet locations to meet the minimum distance requirement, we deployed fewer sheets to ensure sample independence. Once bats settled on their roosting trees, we collected individual fecal samples from each sheet (median 1 sample per sheet, range 0–13), generally within three hours after dawn. We collected approximately four grams of feces per sample and stored them in individual resealable bags and then preserved at a temperature of −20 °C or at −80 °C until laboratory analysis. As flying fox feces exhibit high variability in color and consistency, independent samples can typically be easily identified. We considered each sample collected from a sheet as a sheet-level replicate. During the sample collection process, we recorded the presence of each bat species and the number of bats present on the tree branches above the plastic sheets by direct observation. The field team collected under-roost data using REDCap (Research Electronic Data Capture) software, which exported data into CSV files for analyses.

## Coronavirus screening
**Sample selection and pooling.** We optimized testing effort and costs by applying a two-phase screening approach using pooled samples[33]

(Supplementary Fig. 2). We screened all individual bats for which fecal samples were available, including both BFF and GHFF. For samples from individual bats, RNA was extracted from each sample separately (extraction described below), then we randomly assigned the RNA extract products from each sample into pools of three from within the same sampling session. Depending on the total number of samples obtained in a session, one pool per session might have only consisted of one or two samples. In the second step, if a pool yielded a positive coronavirus detection, the extracts of samples included in the pool were re-analyzed individually. During this second step, we identified the specific individual bat or bats that contributed to the positive result. When pools rendered a negative result, all individual bats that contributed a sample to that pool were categorized as negative (Supplementary Fig. 2).

For fecal samples obtained from the population-level under-roost sampling, we first filtered the available samples to include only those ones that were noted to have BFF roosting over the sheet but no GHFF. Next, for each site and sampling session, we randomly selected 30 samples using the following rules: (1) Using stratified random sampling, one sample was selected from each sheet. Additional samples were then selected from other sheets with the goal of avoiding overrepresentation of a single sheet with a disproportionately large number of samples or creating pools with multiple samples from the same sheet. And (2) after identifying all samples available for a session, a systematic sampling approach was used to allocate them into the pools. This involved arranging the samples by sheet and then the replicate number. The first sample, along with the 11th and 21st, was placed in pool 1, and so forth. The samples within each group were then shuffled to prevent pools composed of samples from identical sheets. Under-roost pools were assigned as positive or negative without rescreening of component samples (Supplementary Fig. 2).

**RNA extraction and coronavirus RT-PCR screening.** Fecal samples were combined with Zymo 1X DNA/RNA shield (0.06 g feces:0.6 ml shield, -1:10) to inactivate the sample, vortexed for 3 min and centrifuged $15,000 \times g/1$ min to clarify, then stored at $-20\,°C$ prior to extraction. RNA was extracted using the MagMAX™ CORE Pathogen kit (Thermo Fisher) and MagMAX automated magnetic particle processor (MagMAX_Core_200 script). Briefly, 200 μl of clarified supernatant was added to 450 μl lysis buffer in a 1.5 ml centrifuge tube, vortexed for 3 min, centrifuged $15,000 \times g/2$ min, then 500 μl of lysate added to 30 μl magnetic bead/protein kinase mix in a deep well plate, sealed and mixed on a plate shaker at high speed for 2 min. To this, 350 μl binding solution was added prior to proceeding with wash and elution steps according to the kit manufacturer's instructions. RNA was then converted to cDNA with the Invitrogen SuperScript IV VILO Master Mix with ezDNase (Thermo Fisher) as per the manufacturer's instructions. We used a semi-nested RT-PCR assay to target the betacoronavirus ORF1b region with the AllTaq PCR Core kit (QIAGEN). The primers for the first round were: VM3007 5′-GGTTGGGAYTAYCCH AARTGYGA-3′; VM3008 5′-CCRTCATCAGAHARWATCATC-3′; VM3009 5′-CCRTCATCACTHARWATCATC-3′. The primers for the second round were: VM3008; VM3009; VM1818 5′-GAYTAYCCHAARTGTGAYAGAGC-3′; VM3010 5′-GAYTAYCCHAARTGTGAYMGHGC-3′[68]. For each PCR, 8 pmol of primer was added per 20 μl reaction with 25 and 40 cycles used for the first and second round, respectively. Known positive samples (Human alpha-CoV NL63 & beta-CoV HKU1) and no template controls were used to confirm assay performance.

**DNA sequencing and assembly.** The expected amplicon, measuring 434 base pairs, was verified by gel electrophoresis before purification using AMPure XP (Beckman Coulter). In addition to individually positive reactions and to potentially capture any lower-yield samples without visual band, we also combined PCR products in equal volumes across the rows of plates for bead purification. Next, the purified DNA

was sequenced on the Illumina iSeq 100 platform using the Nextera XT DNA library preparation kit with unique dual indexes generating at least 100,000 paired reads per sample (2 × 150nt). For data analysis, sequence reads were trimmed of adapter and low-quality bases using BBDuk (available from https://sourceforge.net/projects/bbmap/) before de novo assembly with MEGAHIT[69]. To identify coronavirus sequences and exclude any non-specific hits, contigs were aligned against the NCBI nucleotide and protein databases using BLAST[70].

**Phylogenetic analysis and identification of viral clades.** All coronavirus sequences were first aligned against the Rousettus bat coronavirus HKU9 (GenBank accession NC_009021) using MAFFT[71] to confirm the presence of the expected region and coverage of each amplicon. Any sequences with less than 250nt of coverage across the target region or less than 10% of relative abundance within each library were removed. The final curated dataset was codon aligned against known reference strains available from NCBI GenBank using MAFFT focusing on betacoronaviruses that were exclusively identified in this study. To confirm clusters and identify viral clades, a maximum likelihood phylogeny was prepared using PhyML[72] with the GTR + G substitution model. Branch support values for the ML tree were estimated using the Shimodaira–Hasegawa approximate likelihood-based test with virus clades (clades) defined as monophyletic groups with SH-like support values >0.8 or genetic distance >10% from other sequences across the target region.

**Whole genome sequencing.** To obtain genome sequences, the previously extracted RNA of fecal samples of select representative strains were processed using a previously published metatranscriptomic approach[73]. Samples were selected with priority from: higher viral load samples (i.e. stronger bands on gels); different sampling sessions; individual animals; and no co-detections. The extracts were first treated with ezDNase Enzyme (Thermo Fisher) to remove residual genomic DNA, and then treated with QIAseq FastSelect −5S/ 16S/ 23S and -rRNA HMR (Qiagen) to deplete ribosomal RNA. Next, SuperScript IV VILO Master Mix (Thermo Fisher) was used for first strand cDNA synthesis, followed by addition of Sequenase Version 2.0 DNA Polymerase (Thermo Fisher) for second strand cDNA synthesis. The double stranded cDNA was purified with Mag-Bind TotalPure NGS (Omega Bio-tek) before Nextera XT DNA Library Preparation Kit with Unique Dual Indexes (Illumina) and paired end (2 × 150nt) sequencing on Illumina NovaSeq (targeting >20 M reads per library). Similar to the amplicon assay, sequence reads were first trimmed using BBDuk before de novo assembly with MEGAHIT and coronavirus sequence identification by nucleotide and protein blast searches against the NCBI nt/nr database. Due to the relatively high rates of co-infection/co-detection in this cohort, careful inspection of the final assembled reads was made to ensure no chimeric assemblies were obtained, with this determined based on the uniformity of coverage and lack of heterogeneity amongst the reads in the assembly set.

The final sequences were annotated using the nobecovirus strain AMB130 using Geneious Prime 2023.1 before alignment against all available reference genomes on NCBI Genbank with MAFFT. Recombination in the sequence alignments was assessed using the Recombination Detections Program v4 with default settings[74] and phylogenetic analysis with PhyML with the GTR + G substitution model and SH-like tests for node support. Sequences were classified according to the latest ICTV criteria (https://ictv.global/report/chapter/coronaviridae/coronaviridae). Briefly, an initial BLAST alignment against the Conserved Domain Database[75] of the ORF1ab region was used to identify and extract the coding regions for the 3CLpro, NiRAN, RdRP, ZBD and HEL1 domains. The translated sequences were then concatenated and aligned using MAFFT before phylogenetic analysis using PhyML with the LG + G amino acid substitution model. Distance matrices of the Percentage of Unchanged Differences (PUD)

and Pairwise Patristic Distance (PPD) between viruses were calculated in Geneious Prime. Representative genomes of each coronavirus clade were submitted to NCBI GenBank (Supplementary Table 3).

**Cytochrome b gene sequencing and host species confirmation.** To confirm the species contributing to under-roost pools, we screened RNA/DNA extracts using PCR targeting the cytochrome b mitochondrial gene[76]. Here, we chose to screen the entire column of extraction plates containing any CoV-positive samples that included all CoV positive wells ($n = 72$) and neighbouring non-CoV positive wells ($n = 216$). PCR was performed from the fecal extracts without cDNA synthesis using the Invitrogen Platinum SuperFi II PCR Master Mix (Thermo Fisher) with the primers L14724 (5′-CGAAGCTTGA-TATGAAAAACCATCGTTG-3′) and H15149 (5′-AAACTGCAGCCCCTCA-GAATGATATTTGTCCTCA-3′) as per manufacturer's instructions. All amplicons of approximately the expected size (~425 bp) were sequenced using the Rapid Barcoding Kit 96 and R9.4.1 flowcells on a MinION Mk1C with rapid basecalling (Oxford Nanopore Technologies) targeting ~10,000 reads per samples. Following this, de-multiplexed sequence reads were filtered by length (between 250 and 450 bp) before mapping against a database of human and flying fox reference sequencing in Geneious Primer using Minimap2[77] with a minimum mapping rate of 25 reads. Consensus sequences were generated based on quality scores in Geneious Prime, which first excludes bases with a Phred quality score below 60% of the maximum possible score, then determines a majority consensus from the most frequent base among the remaining high-quality reads. Unmapped reads were de novo assembled in Geneious Prime and compared to NCBI GenBank using BLAST to determine any other species present. We then compared host data obtained through cytochrome b sequencing to the species recorded as roosting over the sheets in field data. Finally, host-species identification from the RNA sequencing data used for CoV WGS was also performed by mapping the filtered reads against a comprehensive database of cytochrome c oxidase I (COXI) gene sequences (available from https://github.com/bachob5/MetaCOXI).

## Statistical analyses

To estimate prevalence dynamics jointly from pooled under-roost samples and individual catching data we developed a Bayesian data integration approach. This statistical method enables the calculation of time-varying population-level prevalence estimates derived from both sample types, which offers a snapshot of the underlying biologically continuous process of infection dynamics within the populations. Moreover, the individual samples and their associated metadata are utilized to discern individual bat-level characteristics related to variations in prevalence and, in turn, environmental and ecological drivers of viral shedding.

Estimating population-level prevalence, or the proportion of individuals actively shedding virus from pooled samples, can be challenging. A positive detection in a pool of samples indicates that at least one individual in the pool tested positive, but on its own is not directly indicative of population prevalence. In addition to individual prevalence, the proportion of pools testing positive also depends on the pool size. Consequently, to obtain reliable estimates of the true population-level prevalence, the pooled prevalence estimates need to be adjusted to account for the pooling process. This adjustment requires transforming the probability of a positive pool to the individual scale.

Prior to collecting data for this study, we examined the statistical properties of integrating pooled and individual samples[33]. For a fixed budget for laboratory testing, we found that allocating laboratory tests toward pools yielded more accurate and more precise estimates of prevalence (with tighter credible intervals) than if those resources were used to screen individual samples. However, the integration of both data types allowed us to leverage complementary strengths: individual-level data provides crucial demographic information, while

combining individual samples with the pooled data enables precise estimates of true prevalence with broader temporal and spatial coverage with reduced cost and effort.

Moreover, the observed prevalence at any given sampling session is influenced by the epidemic momentum within the bat population, which is temporally correlated but unknown at the time of sampling. Thus, our model framework includes a transformation of pooled probabilities, to account for pooling, and a temporally explicit Gaussian Process (GP) to estimate population prevalence over time. By using a GP, our modeling framework can be formulated in a manner analogous to a logistic regression model—but additionally as one that accounts for both temporal correlation and the pooled structure in the data. Specifically, this model framework can be expressed as

$$y_{t,i} \sim \text{Bernoulli}(\pi_{t,i}) \qquad (1)$$

$$\pi_{t,i} = 1 - (1 - p_{t,i})^{m_{t,i}} \qquad (2)$$

$$\text{logit}(p_{t,i}) = X_{t,i}\beta + \omega_t \qquad (3)$$

where $y_{t,i}$ is a binary variable denoting whether the $i^{th}$ pool at time $t$ tests positive, $\pi_{t,i}$ is the probability of pool $i$ testing positive at time t, which is a function of the individual prevalence $p_{t,i}$ and the pool size $m_{t,i}$. The individual level prevalence is a function of covariates ($X_{t,i}$), in the same fashion as with logistic regression with $\beta$, and a time-varying term ($\omega_t$) that comes from the Gaussian process. For all Bayesian analyses we used weakly informative priors and ran four Markov chain Monte Carlo chains for a minimum of 2000 iterations.

**Model selection approach.** Using the model outlined in Eqs. 1–3, we assessed how the dynamics of viral shedding differed across viral clades, time of the year, and geographic locations. We evaluated the support for competing hypotheses representing biological processes by formulating four distinct model frameworks: (1) a unified model that combines all viral clades and locations—representing the hypothesis that all coronaviruses exhibit the same temporal dynamic across all sites; (2) a location-specific model that pools all viral clades together – representing the hypothesis that all coronaviruses exhibit identical temporal dynamics, but with site-specific differences; (3) a clade-specific model that combines all locations together – representing the hypothesis that every viral clade exhibits a unique temporal dynamic, but consistent across all sites; and (4) a clade-by-location model that applies a unique curve for each combination—representing the hypothesis that every viral clade exhibits a distinct temporal dynamic that varies by site. This fourth framework resulted in a set of models including 30 combinations of viral clades and locations (see Supplementary information for details). For every model formulation, we implemented a cross-validation approach to compare their predictive capacities and assess their performance and thus support for the underlying hypotheses. We conducted a comparative analysis of the models by using "leave-one-out" cross-validation and evaluating the expected log pointwise predictive densities (ELPD)[78], or equivalently LOOIC (leave-one-out information criteria).

**Individual level dynamics of infection.** While the pooled data collected from feces under the roosts cannot be associated with individual bats, data collected from individual bats during catching sessions can provide additional information on how age and species impact prevalence dynamics. From the individual samples the age and species of the bats were recorded. Using individual samples only we fit a dynamic binary regression model using the model specified in Eqs. 1–3; however, with pools of size one this reduces to the standard logistic regression framework coupled with a time-varying term from the GP.

With this model framework, differences in prevalence dynamics can be explored across age, sex, and species for each coronavirus clade and evaluated using LOOIC.

**Association across viral clades.** To assess potential co-infection, or statistical associations, between viral clades and bat age classes, we performed a chi-squared test of association. For this test we used only the results from individual bats. We aimed to understand whether an individual flying fox is more likely to be infected with two coronavirus clades if it was already infected with one clade. This analysis can indicate potential co-infection associations. Following this initial step, we estimated the conditional probabilities of individual bats testing positive for a specific viral clade given that they were positive for another one. The combination of these two approaches allowed us to discuss the potential for co-infections and interactions between pairs of viral clades.

### Reporting summary
Further information on research design is available in the Nature Portfolio Reporting Summary linked to this article.

## Data availability
The field data and CoV clade detection data generated and analysed in this study have been deposited in the Cornell University eCommons Digital Repository, available at: https://doi.org/10.7298/w7sw-6161[79]. The combined processed data used as input for models and figures are available at https://zenodo.org/records/15626080[80], with the relevant code. Sequences used and generated in this study are available in online repositories as per links below. Additional details are provided in Supplementary Table 3. Existing GenBank sequences: ON872523, OK067319. New GenBank sequences: PV683367, PV683362, PV683361, PV683360, PV683359, PV683363, PV683365, PV683366, PV683364. New SRA sequences: SRR33676035, SRR33676034, SRR33676033, SRR33676032, SRR33676031, SRR33676030, SRR19790900, SRR33676029, SRR33676028, SRR33676027,

## Code availability
Code generated during the current study, and the input data for the models and figures, are available at: https://zenodo.org/records/15626080[80]. This analysis used version 4.4.1 of R and version 2.32.2 of stan.

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

## Acknowledgements

We acknowledge the Kabi Kabi, Turrbal, Widjabul Wia-bal, Yugambeh and Yuggera Ugarapul people, who are the Traditional Custodians of the land upon which this work was conducted. We would like to thank Peggy Eby for her contributions to study design, ecological insights and comments on an earlier version of this manuscript. We gratefully acknowledge Maureen Kessler, Liam Chirio, Mandy Allonby, Rachael Smethurst, Remy Brooks, Liam McGuire, Kirk Silas, Ticha Padgett-Stewart, Denise Karkkainen, Justine Scaccia, Ariana Ananda, Emma Glennon, Hannah Eiseman, Cinthia Pietromonaco, and many other volunteers for their assistance in the field. We would like to thank Cinthia Pietromonaco, Liam Chirio, and Anna Waldron for assistance with laboratory work. We also thank Emma Spence for help with figures and references, Sara LaTrielle, Isaac Knights, Dian Riseley, and Stella Maris Januario da Silva for project support, and the Parks family and other landholders for kindly granting us access to the property. The authors acknowledge the University of Sydney's high-performance computing cluster Artemis for providing the high-performance computing resources that have contributed to the research results reported within this paper. The project was supported by the National Science Foundation (DEB1716698, EF2133763, EF-2231624) and the DARPA PREEMPT program Cooperative Agreement # D18AC00031. The content of the information does not necessarily reflect the position or the policy of the U.S. government, and no official endorsement should be inferred. AJP was supported by an ARC DECRA fellowship (DE190100710) and a University of Sydney Horizons Fellowship. MRA was supported by the U.S. Forest Service International Programs (Agreement 22-DG-11132762-347), and the work is a contribution of the Forest and Wildlife Research Center, Mississippi State University, supported by McIntire-Stennis funds.

## Author contributions

A.J.P., M.R.-A., E.J.A., K.P., V.J.M., J.-S.E and R.K.P. designed the study and approach. D.N.J.-S., T.J.L., A.S.D., C.A.F. and D.E.C. collected and coordinated the samples. K.K., K.P. and J.-S.E. performed the viral screening, viral sequencing and host sequencing. B.S. and A.H. developed the modelling approach. A.J.P., J.-S.E., M.R.-A., C.A.F., D.E.C. and A.H. analysed the data. M.R.-A., A.H. and R.K.P. wrote the first draft manuscript and A.J.P., J.-S.E., M.R.-A., A.H., and R.K.P. wrote and edited subsequent versions. All authors contributed to the final version of the manuscript.

## Competing interests

The authors declare no competing interests.

## Additional information

¹Centre for Planetary Health and Food Security, Griffith University, Nathan, QLD, Australia. ²Sydney School of Veterinary Science, University of Sydney, Camperdown, NSW, Australia. ³Sydney Infectious Diseases Institute (Sydney ID), Faculty of Medicine and Health, The University of Sydney, Camperdown, NSW, Australia. ⁴Department of Public and Ecosystem Health, Cornell University, Ithaca, NY, USA. ⁵Department of Wildlife, Fisheries, and Aquaculture, Mississippi State University, Starkville, MS, USA. ⁶Centre for Virus Research, Westmead Institute for Medical Research, Westmead, NSW, Australia. ⁷Department of Mathematical Sciences, Montana State University, Bozeman, MT, USA. ⁸Department of Statistical Science, Duke University, Durham, NC, USA. ⁹Laboratory of Virology, Division of Intramural Research, National Institute of Allergy and Infectious Diseases, National Institutes of Health, Hamilton, MT, USA. ¹⁰EquiEpiVet One Health Epidemiology and Veterinary Science, The Oaks, NSW, Australia. ¹¹Department of Microbiology & Cell Biology, Montana State University, Bozeman, MT, USA. ¹²Odum School of Ecology, University of Georgia, Athens, GA, USA. ¹³Center for the Ecology of Infectious Diseases, University of Georgia, Athens, GA, USA. ¹⁴Department of Biological Sciences, Texas Tech University, Lubbock, TX, USA. ¹⁵These authors contributed equally: Alison J. Peel, Manuel Ruiz-Aravena, Karan Kim. ✉e-mail: andrew.hoegh@montana.edu; js.eden@sydney.edu.au; rkp57@cornell.edu

