## [Transparent Peer Review file · Nature Communications]

Synchronized seasonal excretion of multiple coronaviruses coincides with high rates of coinfection in immature bats

Corresponding Author: Professor Raina Plowright

Version 0:

Reviewer comments:

Reviewer #1

(Remarks to the Author)

The article presented by Peel et al. addresses a very important question regarding zoonotic spillover: how the dynamics of bat infection can impact the emergence coronaviruses. Authors have followed the circulation of bat nobecoviruses for 3 years, leading to the conclusion that the presence of juveniles is one of the major drivers of the seasonality observed for this coronavirus.

Despite its importance and novelty, especially considering the prevalence at the clade level and not at the subgenus level, I have some concerns/comments to address:

Major comments:

- I totally agree with authors when they claim that defining a prevalence based on pooled data is biased. So I can't understand the added value of combining data from individual bats with data from pooled bat samples. Did the authors compare the statistical models and corresponding prevalence obtained from individual bats only vs pooled bats only vs individual + pooled bats? The authors should describe the rationale behind this choice of combining both samples and add a section describing the comparison of the two methods. Whatever there is a difference or not, it would add important methodological information to people studying the circulation of coronaviruses in bats.
- The clade definition seems weak, especially knowing the importance of recombination in CoV evolution. To my opinion, I would have at least followed the rules of the ICTV of the definition of a CoV species, subgenus, etc., based on the complete replicase domain and not to restrict to a 400-bp region of the RdRP.
- It is of major importance to measure virus shedding at the individual bat level. Pteropus bats are big enough to carry a GPS. Why do the authors did not choose to proceed to the capture / release / capture method to follow their bat population? It would be much more accurate to monitor prevalence across time at the individual bat level looking at the same individuals.

Minor comments:

- Authors have sequenced the full genome of representative samples for the different clades: how did they select these representative genomes? Knowing that recombination could occur even within clades (as demonstrated by the authors), conclusions regarding recombination events are a bit weak.
- As briefly mentioned in the discussion, co-infection is a very specific term designating the presence of two different viruses with the same cell of the same host. According to this definition, I would prefer describe co-detection instead of co-infection.
- Did the authors identify other viruses in the virome sequencing of representative bats?
- For bats that are co-infected by at least 2 clades, how did the authors deal with genome assembly and the risk of producing chimeric assemblies?
- Did the authors determine the limit of detection of their PCR? Is it possible that bats that were recaptured were indeed positive during the whole follow-up, but undetectable because of a viral load below the limit of detection?
- Introduction lines 62-64: HKU1 (and possibly OC43) seasonal CoVs spilled over from rodents, not bats.
- Figure 4A & 4B: please add statistical differences between age groups.
- Discussion lines 345-354: do the authors have an idea of the antibody decay in bats? Is it possible that bats can be reinfected, even when adults, which could contribute to the pic in prevalence?
- Discussion lines 384-389: authors proposed that the date of sampling is more important than the place of sampling when implementing a surveillance program. I agree, but I'm wondering if this observation can be due to the Pteropus itself that can fly and migrate over long distances? Do the authors think similar hypotheses could be proposed for insectivorous bats, less prone to long flight?

Comments on Materials and Methods section:

- Please add an independent paragraph described the ethical considerations at the beginning of the MM section.
- Lines 425-426 “Site selection criteria included attributes associated with viral spillover risk and feasibility”: can the authors detail these attributes?
- Lines 448-450: role of the PIT tag?
- The sample selection and pooling strategy is not clear, please add a scheme.

(Remarks on code availability)

Reviewer #2

(Remarks to the Author)

General Comments

This study offers a detailed examination of coronavirus dynamics in Pteropus bat populations in Australia, providing important insights into seasonal viral shedding, co-infection rates, and recombination potential.

The results confirm observations from other longitudinal studies, particularly the link between viral shedding and bat age. While the study benefits from a substantial dataset collected over three years and employs a novel Bayesian approach to estimate prevalence dynamics, the findings do not substantially advance previous knowledge in the field. Additionally, the limited bat species diversity restricts the study's broader implications for understanding coronavirus circulation and dynamics across other bat species.

I should note that I do not have the expertise to fully assess the modeling analysis used in this study. Based on my understanding, however, the findings may not meet the criteria for publication in Nature Communications.

Minor Comments

- Provide information on the bat species present at the roosts where under-roost sampling was conducted.
- Revisit individual samples within each positive pool for species identification.
- If feasible, analyze individual fecal samples from the five pools that tested positive for Clade 2d.iii for more precise species identification.

(Remarks on code availability)

Reviewer #3

(Remarks to the Author)

Peel et al. present a large-scale betacoronavirus surveillance study in Australian Pteropus bats. The paper is well written and provides a rich analysis of the collected data leading to some interesting conclusions. The authors clearly show a higher prevalence of the sampled betacoronaviruses in younger bats (that has been implicated in previous research) as well as highlight how different nobecovirus clades have different seasonal patterns, which is quite an important finding for the field. I only have a few minor suggestions which I outline below:

1. The co-roosting patterns of these bats with other species is rather important when it comes to identifying the true host of the detected viruses (especially when it comes to pooled sampling). The authors refer to mitochondrial sequence data but do not present them in the paper. If host sequences are available for the pooled samples I would urge the authors to include these as supplementary in the current manuscript (for example perform some blast-based bat transcript identification analysis for their next-gen sequencing). If such sequencing data isn't available for the pooled samples the authors should expand on field observations about other potential species co-roosting with the Pteropus bats in the text.

2. lines 154-155: I would urge against naming the new open reading frame ORFx. “ORFX” is a named protein in sarbecoviruses (<https://journals.asm.org/doi/10.1128/jvi.03079-15>) and using the same name can lead to confusion, especially since the virus groups are closely related.

3. In table 2 I noticed that the location-specific hypothesis has the worst fit. Would that be explained by movement of individuals between the sampled locations? Do the authors have any evidence of that and how often would the expect movement between locations (and potential transmission of the viruses) to happen?

4. Could the authors ensure that collection date, location (ideally map coordinates) and host species are available for all their submitted virus sequences? Either in the paper or (better) in their genbank entries.

(Remarks on code availability)

Version 1:

Reviewer comments:

Reviewer #1

(Remarks to the Author)

I would like to thank the authors for the huge efforts they have made to answer to the questions I had during the first round of review, even when my questions were more linked to scientific curiosity than a problem with the methodology they employed (for example with bat Ab decay). Especially I appreciate the demonstration of how the integrated approach can estimate more precisely the prevalence by pooling individual and pool positivity frequency. This is of major importance, and I wonder if this method should not be published separately to have a broader diffusion.

(Remarks on code availability)

Reviewer #2

(Remarks to the Author)

Thank you for addressing my comments thoroughly and attentively. I appreciate your thoughtful responses and am satisfied with them.

(Remarks on code availability)

I do not have the expertise to assess the accuracy of the code.

Reviewer #3

(Remarks to the Author)

The authors have addressed all my concerns, I'm happy to recommend this paper for publication.

(Remarks on code availability)

The doi under data availability (<https://doi.org/10.7298/w7sw-6161>) does not seem to be active yet, but the code availability link works and its content is comprehensive.

"Synchronized seasonal excretion of multiple coronaviruses in Australian Pteropus spp is associated with co-infections in juvenile and subadult bats."

RESPONSE TO REVIEWERS

We thank the reviewers for their time, and for providing thoughtful and constructive feedback on our manuscript, which has helped us improve its clarity and rigor. We have addressed all comments and suggestions in our point-by-point response below.

The main changes to the manuscript include:

- *Enhanced analysis and presentation of our sampling framework, including new supplementary figures that (1) more clearly illustrate our sampling schema and pooling strategy, and (2) demonstrate how prevalence estimates differ when using individual samples alone, pooled samples alone, or our integrated approach.*
- *Addition of extensive host cytochrome b sequencing results for pooled samples, with sequence data submitted to GenBank, providing further support for host species identification*
- *Classification of virus sequences according to ICTV criteria, confirming that clades i-v belong to a single species within the Nobecovirus subgenus*
- *Updated annotations for all submitted virus sequences, including comprehensive collection metadata*
- *More detailed discussion of the relationship between bat mobility and viral dynamics, including the implications for surveillance strategies in highly mobile species*

We also note that revision of code identified one small transcription error, which has now been corrected. This identified one additional co-infected individual, but otherwise had negligible effects on calculated prevalences, and no impact on modeling output or interpretations.

All changes are marked in the revised manuscript using tracked changes. Line numbers in our responses refer to the track change version of the revised manuscript.

Reviewer #1

Remarks to the Author: The article presented by Peel et al. addresses a very important question regarding zoonotic spillover: how the dynamics of bat infection can impact the emergence of coronaviruses. Authors have followed the circulation of bat nobecoviruses for 3 years, leading to the conclusion that the presence of juveniles is one of the major drivers of the seasonality observed for this coronavirus.

We thank the reviewer for their positive assessment of the importance of this work.

Despite its importance and novelty, especially considering the prevalence at the clade level and not at the subgenus level, I have some concerns/comments to address:

Major comments:

- I totally agree with authors when they claim that defining a prevalence based on pooled data is biased. So I can't understand the added value of combining data from individual

bats with data from pooled bat samples. Did the authors compare the statistical models and corresponding prevalence obtained from individual bats only vs pooled bats only vs individual + pooled bats? The authors should describe the rationale behind this choice of combining both samples and add a section describing the comparison of the two methods. Whatever there is a difference or not, it would add important methodological information to people studying the circulation of coronaviruses in bats.

We appreciate the opportunity to clarify our rationale and acknowledge that the benefits of our proposed statistical approach need to be better described in the manuscript.

We agree that estimates of prevalence (the proportion of individuals in the population that are shedding the virus) from pooled data would be biased if calculated as the proportion of positive pools. However, using Equations 1 & 2, this bias can be corrected, to get accurate estimates of prevalence. Prior to collecting data for this study, we examined the statistical properties of integrating pooled and individual samples (Hoegh et al. 2021, #34 in reference list). For a fixed budget for laboratory testing, we found that allocating laboratory tests toward pools yielded more accurate and more precise estimates of prevalence (with tighter credible intervals) than if those resources were used to screen individual samples. However, the integration of both data types allowed us to leverage complementary strengths: individual-level data provides crucial demographic information, while combining individual samples with the pooled data enables precise estimates of true prevalence with broader temporal and spatial coverage with reduced cost and effort.

To emphasize this point, we have re-analyzed the data to demonstrate how the prevalence curves from Figure 2 would differ when using individual samples alone (Fig R1a, below), pooled samples alone (Fig R1b), or the integrated approach we originally implemented in the manuscript (Fig R1c). The most apparent difference in these figures is the wider credible intervals in Fig R1a and Fig R1b, which correspond to time periods where fewer samples of that type were tested. By combining both sample types, we achieved more precise credible intervals, as shown in Fig R1c. Another notable point of comparison on these figures is during March 2019 for clade 2d.iv. The combined result with both individuals and pools (Fig R1c), results in a higher prevalence estimate than would be derived from the pools alone, while providing a more precise interval than either individuals (Fig R1a) or pools (Fig R1b) alone. While the “true prevalence” in this population remains unknown, we can infer from our previous simulation results that this combined prevalence estimate more closely represents true prevalence than either individual or pooled estimates alone.

To clarify this in the manuscript, we have added:

- *[Introduction] lines 103-104: “This approach combined the efficiency of pooled sampling with the inferential capabilities of estimation from metadata from individuals, and results in more accurate and precise estimates of prevalence than either sampling approach alone.”*
- *[Results] line 227: reference to a new supplementary figure, which compares the results from the individual, pooled and combined dataset results, and an associated explanation, as above.*
- *[Methods] lines 793-800: “Prior to collecting data for this study, we examined the statistical properties of integrating pooled and individual samples³⁴. For a fixed budget for laboratory testing, we found that allocating laboratory tests toward pools yielded more accurate and more precise estimates of prevalence (with tighter credible intervals) than if those resources were used to screen individual samples. However, the integration of both data types allowed us to leverage complementary strengths: individual-level data provides crucial demographic information, while combining individual samples with the pooled data enables precise estimates of true prevalence with broader temporal and spatial coverage with reduced cost and effort.”*
- The clade definition seems weak, especially knowing the importance of recombination in CoV evolution. To my opinion, I would have at least followed the rules of the ICTV of the definition of a CoV species, subgenus, etc., based on the complete replicase domain and not to restrict to a 400-bo region of the RdRP

Our general take was to identify well-supported phylogenetic groups (clades/lineages - defined as monophyletic groups), and in this case, the partial RdRP phylogenies demonstrated there were six distinct clades. Once we performed the whole genome analysis, the same groupings held, except for regions of inter-clade recombination as highlighted in the paper.

However, to address this point more specifically, we have used our genome data to classify the sequences according to the latest ICTV criteria: <https://ictv.global/report/chapter/coronaviridae/coronaviridae> that uses the 3CLpro, NiRAN, RdRP, ZBD and HEL1 domain (ORF1ab) amino acid sequences to look at sequence identity and branch lengths to define genera, subgenera and species. With this additional analysis, we show that the main five clades we identified (excluding the HKU9 2d.vi virus) would be part of a single species within the nobecovirus genus but remain monophyletic in this same region (which is essentially an expanded RdRP region), so we have kept using the term ‘clade’ below the level of more formal classifications.

To clarify this, we have added the following text in the results section, in lines 188-193:

“To further refine the classification of the viruses identified here, we analysed conserved replication genes (3CLpro, NiRAN, RdRP, ZBD, and HEL1) following ICTV criteria, which suggested that clades i-v were together members of a single species within the Nobecovirus subgenus, and distinct species from both AMB130 and HKU9.”

And further detail in the method, in lines 740-748:

Sequences were classified according to the latest ICTV criteria (<https://ictv.global/report/chapter/coronaviridae/coronaviridae>). Briefly, an initial BLAST alignment against the Conserved Domain Database (ref) of the ORF1ab region was used to identify and extract the coding regions for the 3CLpro, NiRAN, RdRP, ZBD and HEL1 domains. The translated sequences were then concatenated and aligned using MAFFT before phylogenetic analysis using PhyML with the LG+G amino acid substitution model. Distance matrices of the Percentage of Unchanged Differences (PUD) and Pairwise Patristic Distance (PPD) between viruses were calculated in Geneious Prime.

- *It is of major importance to measure virus shedding at the individual bat level. Pteropus bats are big enough to carry a GPS. Why do the authors did not choose to proceed to the capture / release / capture method to follow their bat population? It would be much more accurate to monitor prevalence across time at the individual bat level looking at the same individuals.*

We wholeheartedly agree that it is of major importance to measure virus shedding at the individual bat level and that it would be ideal to monitor prevalence by following the same individuals through time. However, while we agree that GPS tracking could provide valuable movement data, unfortunately there are significant practical and methodological limitations to using GPS tracking to obtain longitudinal data from individuals.

- *Flying foxes are indeed large enough to carry GPS tags, and our team did undertake tracking on a small number of individuals within this study. However, studies that have tracked flying foxes in Australia over long periods (attaching the GPS tag with a collar) have been associated with significant adverse welfare issues in a subset of individuals. For this reason, our group attached trackers with glue, which lasts around 10-30 days, limiting the ability for long-term tracking over a longitudinal study period.*
- *Additionally, the high cost per unit (hundreds of dollars, to thousands of dollars) restricts sample size, making it impossible to get unbiased population level estimates.*
- *More importantly, however, flying foxes are highly mobile animals with rapid roost turnover rates and are capable of traveling >100km in a night and thousands of kilometers over consecutive days (Welbergen et al. 2020; Fleming and Eby, 2003; references #65 and #66 in revised manuscript).*
- *However, even with location data it is not possible to recapture specific individuals as these are canopy dwelling bats (not cave-dwelling) aggregating in the thousands to tens of thousands of animals.*
- *Finally, even if individuals could be tracked in real time across these distances, and could be identified amongst a roost of thousands or tens of thousands of bats, the density of individual bats roosts also makes targeted under roost sampling unfeasible.*

We have spent many hours brainstorming on how we could undertake work like this, but currently, it is not possible to monitor prevalence across time at the individual bat level in these species by looking at the same individuals. In the absence of practical and financial constraints, we would sample all of the thousands of individual bats present within a roost at a given time.

However, this is also not feasible. Instead, our study design sought to optimize the estimates of prevalence through systematic temporal sampling across the population while integrating the detailed individual-level data from serial cross-sectional sampling. As we've shown in our response above, within a given testing budget, this approach improves the estimates of prevalence than either approach alone.

We aimed to clarify this through the additions noted in our previous response comparing estimates from individual samples alone, pooled samples alone or our integrated approach (in particular, to the Methods in lines 793-800).

Minor comments:

- Authors have sequenced the full genome of representative samples for the different clades: how did they select these representative genomes? Knowing that recombination could occur even within clades (as demonstrated by the authors), conclusions regarding recombination events are a bit weak.

We originally aimed to simply get two representative genomes for each partial RdRP clade. The initial work focused on the under-roost samples since this was available first with a priority for: higher viral load samples (stronger bands on gels); different sampling sessions; and no co-detections. It wasn't always possible to meet these criteria, and we ended up attempting sequencing with some redundancy (often five samples per clade) and with the hope of getting genome-level coverage from at least two of the samples for each clade. This was how clades 2d.iv-vi were derived. Later, we could use the individual-level samples that were mono-infected to complete the remaining work including clades 2d.i-iii (and re-visit clades that failed to meet previous criteria). In the end, we attempted RNA sequencing on more than 30 samples and met our ideal selection criteria for more than half the genomes. While the remaining included some pooled under-roost samples, the final genomes included were all clearly derived from a single virus source as determined by close inspection of the assembled reads and lack of heterogeneity as would be inspected from mixed infections.

Full details of the genome sequences, including how they were derived, has been added to a new SI Table 3.

We have also clarified this in the methods, see lines 717-718:

“Samples were selected with priority from: higher viral load samples (i.e. stronger bands on gels); different sampling sessions; individual animals; and no co-detections.”

Considering how frequently coronaviruses are known to recombine, our conclusions are fairly conservative, and we only chose to highlight one specific example of inter-clade recombination. See lines 208-217:

“This suggests a possible role for recombination in the diversification of the viruses, and while recombination analysis did propose possible recombination events near the junctions of the coding regions (ORF1a/b, ORF1b/spike and Nucleocapsid/NS7a; SI Table 4), it is somewhat difficult to resolve if these are true recombination events or other

evolutionary effects shaping the trajectory of the major clades. However, we did identify more well-supported circulating recombinant forms including a 2d.ii virus ACRED010_53 (from a juvenile BFF) that has most of its sequence being wild type 2d.ii, with a mosaic insertion of the spike N-terminal domain from a 2d.i virus (SI Figure 8), confirming recombination between the major nobecovirus clades.”

- As briefly mentioned in the discussion, co-infection is a very specific term designing the presence of two different viruses with the same cell of the same host. According to this definition, I would prefer describe co-detection instead of co-infection.

While we appreciate that co-infection has this very specific meaning in virology, it is widely used to describe the presence of multiple infectious agents within an individual host, regardless of cellular co-localization. This usage expands across the fields of disease ecology (Susi et al, Nat Commun, 2015; de Roode et al, Am Nat, 2005), epidemiology (Bal et al, Nat Commun, 2022), genomics (Wang et al, Nat Commun, 2023), veterinary science (Middlebrook et al, Zoonoses Public Health, 2022), human medicine (de Bruyn et al, Nat Commun, 2023) and immunology (Kang et al, Nat Commun, 2022) – published in Nature communications and other high-profile journals. This broader usage appears extensively in the scientific literature studying multiple pathogens in wildlife populations (Hoarau et al., PLoS Path, 2020), including numerous studies of viral infections in bats (reviewed in Smith et al, Viruses, 2023). Given that Nature Communications is a multi-disciplinary journal, we have kept the term co-infection as we feel that it is used in a way that is consistent with existing literature across broad disciplinary readership.

Susi, H., Barrès, B., Vale, P. F., & Laine, A.-L. (2015). Co-infection alters population dynamics of infectious disease. *Nature Communications*, 6(1). <https://doi.org/10.1038/ncomms6975>

de Roode, J., Helinski, M., Anwar, M., & Read, A. (2005). Dynamics of Multiple Infection and Within-Host Competition in Genetically Diverse Malaria Infections. *The American Naturalist*, 166(5), 531-542. <https://doi.org/10.1086/491659>

Bal, A., Simon, B., Destras, G., Chalvignac, R., Semanas, Q., Oblette, A., Quéromès, G., Fanget, R., Regue, H., Morfin, F., Valette, M., Lina, B., & Josset, L. (2022). Detection and prevalence of SARS-CoV-2 co-infections during the Omicron variant circulation in France. *Nature Communications*, 13(1). <https://doi.org/10.1038/s41467-022-33910-9>

Wang, J., Pan, Y.-fei, Yang, L.-fen, Yang, W.- hong, Lv, K., Luo, C.- ming, Wang, J., Kuang, G.- peng, Wu, W.- chen, Gou, Q.-yu, Xin, G.- yang, Li, B., Luo, H.- le, Chen, S., Shu, Y.- long, Guo, D., Gao, Z.-H., Liang, G., Li, J., ... Shi, M. (2023). Individual bat virome analysis reveals co-infection and spillover among bats and virus zoonotic potential. *Nature Communications*, 14(1). <https://doi.org/10.1038/s41467-023-39835-1>

Middlebrook EA, Romero AT, Bett B, Nthiwa D, Oyola SO, Fair JM, Bartlow AW. Identification and distribution of pathogens coinfecting with *Brucella* spp., *Coxiella burnetii* and Rift Valley fever virus in humans, livestock and wildlife. *Zoonoses Public Health*. 2022 May;69(3):175-194. doi: 10.1111/zph.12905. Epub 2022 Jan 15. PMID: 35034427; PMCID: PMC9303618.

du Bruyn, E., Stek, C., Daroowala, R., Said-Hartley, Q., Hsiao, M., Schafer, G., Goliath, R. T., Abrahams, F., Jackson, A., Wasserman, S., Allwood, B. W., Davis, A. G., Lai, R. P.-J., Coussens, A. K., Wilkinson, K. A., de Vries, J., Tiffin, N., Cerrone, M., ... Ntusi, N. A. B. (2023). Effects of tuberculosis and/or HIV-1 infection on COVID-19 presentation and immune response in Africa. *Nature Communications*, 14(1). <https://doi.org/10.1038/s41467-022-35689-1>

Kang, T.G., Kwon, K.W., Kim, K. et al. Viral coinfection promotes tuberculosis immunopathogenesis by type I IFN signaling-dependent impediment of Th1 cell pulmonary influx. *Nat Commun* 13, 3155 (2022). <https://doi.org/10.1038/s41467-022-30914-3>

Hoarau, A. O. G., Mavingui, P., & Lebarbenchon, C. (2020). Coinfections in wildlife: Focus on a neglected aspect of infectious disease epidemiology. *PLOS Pathogens*, 16(9), e1008790. <https://doi.org/10.1371/journal.ppat.1008790>

Jones, B. D., Kaufman, E. J., & Peel, A. J. (2023). Viral Co-Infection in Bats: A Systematic Review. *Viruses*, 15(9), 1860. <https://doi.org/10.3390/v15091860>

- Did the authors identify other viruses in the virome sequencing of representative bats?

Yes, other viruses (plant, insect and mammalian) were present in the feces; however, our use of RNA sequencing was not aimed at describing the viromes of these bats but rather one of the few methods of recovering the genomes of these novel coronaviruses initially identified by the pan-CoV RT-PCR. We have more extensive follow-up work that looks at the viromes more widely and across sample types, so have chosen not to include this detail here. We also feel it is not directly relevant to the aims of this coronavirus-focused study.

- For bats that are co-infected by at least 2 clades, how did the authors deal with genome assembly and the risk of producing chimeric assemblies?

As mentioned above, besides picking individual, mono-infections where possible, we also manually inspected each assembly to check for heterogeneity that might suggest chimeric assemblies. We have clarified our approach in the methods section, see lines 729-732:

“Due to the relatively high rates of co-infection/co-detection in this cohort, careful inspection of the final assembled reads was made to ensure no chimeric assemblies were obtained, with this determined based on the uniformity of coverage and lack of heterogeneity amongst the reads in the assembly set.”

- Did the authors determine the limit of detection of their PCR? Is it possible that bats that were recaptured were indeed positive during the whole follow-up, but undetectable because of a viral load below the limit of detection

The pan-CoV RT-PCR we used was taken from a recently published (<https://pmc.ncbi.nlm.nih.gov/articles/PMC8067199/>) update to the well-characterized primers by Watanbe et al (<https://pubmed.ncbi.nlm.nih.gov/20678314/>). These target all four CoV genera and were slightly more sensitive than the Watanbe primers primarily due to the semi-nested approach rather than just a single round of PCR. In this study, the sensitivity was found to be 1-68 copies/ μ L of RNA. We did not specifically re-test the sensitivity of the primers in our hands and given our work used stools over cultures, the sensitivity would likely be reduced, and that the overall prevalence would likely be underestimated due to false negatives.

We have highlighted this as a limitation in the Discussion in lines 508-511:

“While the use of a pan-CoV RT-PCR enables the detection of a wide diversity of strains, these approaches can lack some sensitivity compared to less degenerate primer sets, and consequently, our estimates of prevalence and re-infection rates might be reduced by false negatives.”

- Introduction lines 62-64: HKU1 (and possibly OC43) seasonal CoVs spilled over from rodents, not bats.

Yes, we agree. As summarised below, most (5 out of 7) human coronaviruses originated from viruses in bats, including:

Two endemic human CoV:

1. human coronavirus 229E(HCoV-229E)
2. HCoV-NL63

And three recently emerged/highly pathogenic human CoV:

3. SARS-CoV
4. MERS-CoV
5. SARS-CoV-2

The remaining two endemic human CoV originated in other species:

6. HCoV-OC43
7. HCoV-HKU1

We assume that the structure of the sentence made this unclear, and so have tried to clarify this with the following minor change:

Lines 62-66: “Genomic studies have established that most human coronaviruses originated from viruses in bats, including two endemic human coronaviruses causing mild respiratory and gastrointestinal infections and the three recently emerged coronaviruses that have triggered devastating outbreaks and pandemics (SARS-CoV, MERS-CoV, and SARS-CoV-2).”

- Figure 4A & 4B: please add statistical differences between age groups.

Thank you for identifying this figure and encouraging us to think carefully about the best way to share the information. We acknowledge the confusion caused by the bar chart and uncertainty intervals that were not formally discussed in the caption. The original intent of this figure was not to formally test for statistical differences between the age groups, but rather summarize the descriptive statistics associated with overall prevalence. If the intent was to conduct formal testing for differences across species and clades, this would require controlling for type I errors due to the multiple comparison, both in accordance with Nature guidelines and good statistical practice. Furthermore, these figures don't account for the temporal sampling of the data, which we know is an important factor in prevalence. Rather, we present this information on Fig 4A & 4B as descriptive statistics according to the Nature guidelines, with “clearly labelled measure of centre (such as the mean or the median), and a clearly labelled measure of variability (such as standard deviation or range).”

Accordingly, we have updated Figure 4A & 4B along with the caption to clarify our intent with this figure and how readers can interpret the information presented here. See below:

“A-B: Empirical prevalence of the six coronavirus clades in individual bats, by species (BFF and GHFF) and age class, is indicated with white dots and text. The uncertainty bars represent 95% Bayesian credible intervals using a uniform prior (Beta(1,1)) on prevalence. Note the different x-axis scale and large uncertainty levels, associated with small sample sizes for GHFF, particularly for adults and subadults, corresponding to the higher prevalence levels (sample sizes for BFF: 172 juvenile/105 subadult/822 adults; GHFF: 22/3/4). These figures are intended to present descriptive summaries of prevalence rather than for determining differences through statistical testing which would require formally controlling for testing multiple hypotheses.”

- Discussion lines 345-354: do the authors have an idea of the antibody decay in bats? Is it possible that bats can be reinfected, even when adults, which could contribute to the pic in prevalence?

*Currently there is very limited data available on the general antibody kinetics in bat species. However, available data suggest a robust induction of a humoral response, followed by a relatively rapid decay. This observation has been shown in experimental studies in big brown bats (*Eptesicus fuscus*) with rabies virus (Turmelle et al., 2010), as well as with Egyptian fruit bats (*Rousettus aegyptiacus*) experimentally infected with Marburg virus (Schuh et al., 2017). These experimental results have been supported by field studies in vampire bats (*Desmodus rotundus*) in Peru, in which observations of significant antibody waning led researchers to propose that rabies maintenance in bats was facilitated by waning of protective immunity (Meza et al., 2022).*

Reinfection could occur following waned immunity over time. However, because the initial infection and immune response likely generated a T and B memory response, the animal will more rapidly mount an adaptive response (the so-called recall response) to the second (or third etc.) infection. This typically results in subsequent infections having both lower magnitude (less virus shed) and shorter duration of shedding.

However, one needs to be cautious in trying to translate these observations across different systems, due to the likelihood of different host-pathogen interactions. The humoral response in bats is typically measured by systemic IgG responses (within sera collected from the bats). In the case of the gastrointestinal coronaviruses, however, the adaptive responses are likely driven by the gut-associated lymphoid tissue and secretory IgA (rather than IgG). While no bat studies have yet focused on mucosal immunity or secretory IgA, these responses likely play a crucial role in both infection and virus shedding patterns.

This is a complex area, and it was out of scope to develop the proteins and assays required to assess bat immune responses to the coronaviruses detected within this study. Acknowledging word count limitations within the manuscript, we have summarised the issues above with the following addition (lines 514-516):

“Finally, we lacked tools for assessing immunity to coronaviruses in bats and serological assays for the specific circulating coronaviruses have not yet been developed, limiting our ability to test for evidence of waning immunity and re-infection.”

We thank the reviewer for raising this as an interesting avenue for future research.

*Turmelle, A.S., Jackson, F.R., Green, D., McCracken, G.F. & Rupprecht, C.E. (2010). Host immunity to repeated rabies virus infection in big brown bats. *J Gen Virology* 91, 2360-2366.*

*Schuh, A.J., Amman, B.R., Sealy, T.K., Spengler, J.R., Nichol, S.T. & Towner, J.S. (2017). Egyptian rousette bats maintain long-term protective immunity against Marburg virus infection despite diminished antibody levels. *Sci Rep-uk* 7, 535.*

*Meza, D.K., Mollentze, N., Broos, A., Tello, C., Valderrama, W., Recuenco, S., Carrera, J.E., Shiva, C., Falcon, N., Viana, M. & Streicker, D.G. (2022). Ecological determinants of rabies virus dynamics in vampire bats and spillover to livestock. *Proc. R. Soc. B* 289, 20220860.*

- Discussion lines 384-389: authors proposed that the date of sampling is more important than the place of sampling when implementing a surveillance program. I agree, but I'm wondering if this observation can be due to the Pteropus itself that can fly and migrate over long distances? Do the authors think similar hypotheses could be proposed for insectivorous bats, less prone to long flight?

Yes, the reviewer is correct. The high mobility of our study species is a critical component to this finding. We have edited the text to address this point (Lines 488-493).

"While spatial sampling remains crucial for capturing potential clade differences across broad geographic regions and habitat types^{18,58} at the scale of our study area (>11,000 km²), we found that temporal surveillance rather than spatial surveillance was more informative for understanding coronavirus dynamics. This finding may be explained, in part, by the extreme mobility of Pteropus bats⁶⁵, and is therefore likely generalizable to other bat species known to travel long distances (such as annually migrating insectivorous bats)⁶⁶."

Comments on Materials and Methods section:

- Please add an independent paragraph described the ethical considerations at the beginning of the MM section.

The following text has been added (Lines 534-537).

"Bat capture and sampling was performed following best practices⁶⁸. Field protocols were approved by the Montana State University Institutional Animal Care and Use Committee (201750) and Griffith University Animal Ethics Committee (ENV/10/16/AEC and ENV/07/20/AEC).

- Lines 425-426 "Site selection criteria included attributes associated with viral spillover risk and feasibility": can the authors detail these attributes?

The following text has been added to clarify this (Lines 544-548):

"Site selection criteria included attributes associated with viral spillover risk and feasibility, including: continuous occupation by BFF, recently overwintering, limited native winter food, and sampling feasibility, access, and permissions³²."

- Lines 448-450: role of the PIT tag?

The role of the PIT tag was to enable identification of recaptured individuals. The following text has been added (Lines 571-572):

"All bats were checked for a Passive Integrated Transponder (PIT) tag to identify recaptured individuals. If no PIT tag was present, we inserted a PIT tag (RFID, ZD Tech Group China) under the skin between the scapulae while the bat was anesthetized."

- The sample selection and pooling strategy is not clear, please add a scheme.

We have added a new supplementary figure illustrating our sampling and pooling strategy (SI Figure 2) with the following caption:

“SI Figure 2. Model-guided screening framework. Individual bat samples: RNA was extracted from each fecal sample separately, then extracts pooled in groups of three within the same sampling session for coronavirus PCR. For individual bat pools testing PCR-positive, component samples were rescreened individually to identify specific positive individuals, and subsequently sequenced to identify clades. B) Under-roost samples: From sheets where only black flying foxes were recorded as roosting overhead, 30 samples were randomly selected and pooled to optimize cost efficiency for extraction (three samples per pool). Samples within pools were shuffled to prevent pools containing multiple samples from identical sheets. Under-roost pool extracts were screened using the coronavirus PCR and, following our model-guided approach to maximize information gain relative to cost, classified as positive or negative without component rescreening. All positive pools were sequenced to determine clades.”

We have added references to this figure in the Results (Line 111):

“Following our model-guided screening framework (SI Figure 2), fecal samples were collected for viral screening directly from individual bats (SI Figure 3, SI Table 1) and through population-level sampling by collecting excreta on plastic sheets placed under the roosts (“under-roost” sampling) (SI Figure 4, SI Table 2).

And as a reference to the figure in multiple locations within the Methods:

Line 610: “We optimized testing effort and costs by applying a two-phase screening approach using pooled samples³⁴ (SI Figure 2).”

Lines 643-644: “When pools rendered a negative result, all individual bats that contributed a sample to that pool were categorized as negative (SI Figure 2).”

Lines 656-657: “Under-roost pools were assigned as positive or negative without rescreening of component samples (SI Figure 2).”

Reviewer #2

Remarks to the Author: This study offers a detailed examination of coronavirus dynamics in Pteropus bat populations in Australia, providing important insights into seasonal viral shedding, co-infection rates, and recombination potential.

The results confirm observations from other longitudinal studies, particularly the link between viral shedding and bat age. While the study benefits from a substantial dataset collected over three years and employs a novel Bayesian approach to estimate prevalence dynamics, the findings do not substantially advance previous knowledge in the field. Additionally, the limited bat species diversity restricts the study's broader implications for understanding coronavirus circulation and dynamics across other bat species.

I should note that I do not have the expertise to fully assess the modeling analysis used in this study. Based on my understanding, however, the findings may not meet the criteria for publication in Nature Communications.

We thank the reviewer for their assessment of our work, and positive reflections on the important insights within our manuscript and our novel Bayesian approach to estimating prevalence dynamics. We respectfully disagree that our target species represents a limitation and the suggestion that our findings do not meet the criteria for publication in Nature Communications.

Below, we have outlined Nature Communications' aims, scope and key criteria for publication and identified the ways in which our manuscript meets these criteria:

- *“Nature Communications is an open access, **multidisciplinary** journal dedicated to publishing **high-quality research** in all areas of the biological, health, physical, chemical, Earth, social, mathematical, applied, and engineering sciences. Papers published by the journal aim to represent **important advances of significance to specialists within each field.**”*
 - *Multidisciplinary:*
 - *Our manuscript incorporates insights, methods and approaches across the field ecology, disease ecology, viral genomics, and data science.*
 - *High-quality research:*
 - *Rigorous longitudinal sampling design of both populations and individuals over three years, unprecedented in its scale. The sensitivity and scale of our approach enabled us to confidently identify both dominant and rare cryptic clades of coronaviruses, some with prevalence below 1%.*
 - *Robust statistical analysis using state-of-the-art methods*
 - *Comprehensive integration of epidemiological and genomic data, allowing us to infer the dynamics and evolution of coronaviruses within bat reservoir host populations.*
 - *Important advances of significance to specialists within each field:*
 - *Development of a novel statistical framework for integrating individual and population-level surveillance data, resulting in increased accuracy and precision of prevalence estimates.*

- *First demonstration of synchronous shedding pulses of bat coronaviruses across multiple different clades, paired with co-infections and evidence of contemporary recombination of dominant and rare clades.*
- *Identification of specific conditions driving viral co-infections and recombination opportunities*

Additionally, to address the reviewer's comment about limited bat species diversity in our study, we note that while broad taxonomic sampling is valuable, our study prioritizes detailed temporal and spatial sampling to provide in-depth understanding of viral dynamics - an understudied gap in the literature compared to snapshot sampling across multiple species.

When expanding across larger numbers of species, methodological compromises are often necessary. Multi-species papers often combine results across species for analyses (e.g. Chidoti 2022, Joffrin 2020, Suu-Ire 2022, Nziza 2020), in order to achieve sufficient statistical power to make robust inferences, despite evidence of species-specific differences in shedding (e.g., Djomsi 2023, Suu-Ire 2022). Attempting to sample both populations and individuals longitudinally over multiple years across all bat species in an area requires prohibitively large sample sizes to make species-specific inferences on viral dynamics. Instead, our study focused on robust temporal inferences from two closely related species.

*Additionally, the few field studies undertaking systematic longitudinal sampling and coronavirus screening over multiple years are predominantly in Africa (Meyer et al. 2024, Geldenhuys et al. 2023, Kettenburg 2022), with none to our knowledge in Southeast Asia or Oceania (Ruiz-Aravena et al., 2021). Our work on *Pteropus alecto* and *Pteropus poliocephalus* is significant as *Pteropus* species represent 19% of all bat species in Oceania and comprise 36% of all *Pteropodid* species in the broader region that also encompasses South and Southeast Asia. Our intensive sampling of these ecologically important species in their core range provides regionally-significant insights into viral dynamics that would not be possible through broader but shallower taxonomic sampling.*

*Geldenhuys, M., Ross, N., Dietrich, M., Vries, J.L. de, Mortlock, M., Epstein, J.H., Weyer, J., Pawęska, J.T. & Markotter, W. (2023). Viral maintenance and excretion dynamics of coronaviruses within an Egyptian rousette fruit bat maternal colony: considerations for spillover. *Sci. Rep.* 13, 15829.*

*Kettenburg, G., Kistler, A., Ranaivoson, H.C., Ahyong, V., Andrianiaina, A., Andry, S., DeRisi, J.L., Gentles, A., Raharinosy, V., Randriambolamanantsoa, T.H., Ravelomanantsoa, N.A.F., Tato, C.M., Dussart, P., Heraud, J.-M. & Brook, C.E. (2022). Full Genome Nobecovirus Sequences From Malagasy Fruit Bats Define a Unique Evolutionary History for This Coronavirus Clade. *Front. Public Heal.* 10, 786060.*

*Meyer, M., Melville, D.W., Baldwin, H.J., Wilhelm, K., Nkrumah, E.E., Badu, E.K., Oppong, S.K., Schwensow, N., Stow, A., Vallo, P., Corman, V.M., Tschapka, M., Drosten, C. & Sommer, S. (2024). Bat species assemblage predicts coronavirus prevalence. *Nat. Commun.* 15, 2887.*

*Ruiz-Aravena, M., McKee, C., Gamble, A., Lunn, T., Morris, A., Snedden, C.E., Yinda, C.K., Port, J.R., Buchholz, D.W., Yeo, Y.Y., Faust, C., Jax, E., Dee, L., Jones, D.N., Kessler, M.K., Falvo, C., Crowley, D., Bharti, N., Brook, C.E., Aguilar, H.C., Peel, A.J., Restif, O., Schountz, T., Parrish, C.R., Gurley, E.S., Lloyd-Smith, J.O., Hudson, P.J., Munster, V.J. & Plowright, R.K. (2021). Ecology, evolution and spillover of coronaviruses from bats. *Nat Rev Microbiol* 20, 299–314.*

Minor Comments:

- Provide information on the bat species present at the roosts where under-roost sampling was conducted.

This information has been added to SI Table 2.

- **Revisit individual samples within each positive pool for species identification.**

Identifying species from component samples within each positive pool would be interesting, however, this was not feasible within our study design. In summary, a key objective of the paper was to use model-guided sampling frameworks to optimize the information gained within a given budget. In turn, this efficient approach helped maximise the spatial and temporal scope of the study, whereby the metadata from individual bats enabled us to explore drivers of viral shedding, while integrating the additional samples from the pooled sampling helped improve overall estimates of prevalence, without needing to screen all component samples individually. We have now added a sampling schema as supplementary figure 2 to clarify this approach.

For the samples from individual bats, individual level data was desirable, so we did re-screen all component samples from CoV positive pools. This allowed us to identify both the CoV-positive individuals and their species (by physical examination), enabling our original analysis of host drivers of viral shedding. However, for under-roost samples, our approach was different. Our analysis showed that the most efficient screening protocol (in terms of information gained versus cost expended) did not include rescreening under-roost pools. For this reason, these samples were pooled during the initial extraction step. Molecular identification species from positive pools would require re-extracting each component sample individually - a costly and time-consuming process that would counteract the efficiency principles of our sampling approach. In fact, many samples have insufficient original sample remaining to undertake this task.

Instead, we have now undertaken extensive host cytochrome b sequencing in a way that makes efforts to address the reviewer's request whilst aligning with our sampling framework. To obtain genetic data of the bat species present in the under-roost samples, we have screened all CoV-positive under-roost pools (n=72) and a major sample of CoV-negative under-roost pools (n=216) using high-throughput sequencing of the cytochrome b gene. Amplification and sequencing were successful for 70% (n=201/288) of the samples tested. Overall, the cohort was dominated by black flying foxes (BFF) with 98.5% of under-roost samples screened (n=198/201), which is consistent with our field observations and our sample selection approach, which targeted samples where BFF were noted as roosting above the sheets. Grey-headed flying fox (GHFF) DNA was detected in eleven pools. For three of these, GHFF DNA was the only host DNA detected, but all of these pools were negative for CoVs. For the remaining eight pools, both GHFF and BFF DNA was detected within the pool, including pools positive for clade 2d.v (n=2), clade 2d.iii (n = 1), 2d.iv (n = 1), and one pool positive for both 2d.iii and 2d.v.

We have incorporated these host PCR and sequencing efforts into both the results and methods:

See lines 131-141:

“Host mitochondrial (cytochrome b) sequences were then amplified for all CoV positive under-roost pools (n=72) and a majority subset of the remaining CoV-negative pools (n=216). Of the 201 under-roost pools that were successfully amplified and sequenced,

BFF was the dominant species identified (98.5%, n=198/201), with a similar rate between CoV-positive and negative pools. Consistent with our individual captures, field observations, and sample selection approach, these results confirm that our study cohort largely comprised BFF. Detection of GHFF DNA was rare in the under-roost pools (5.5%, n=11/201) and GHFF DNA was mostly co-detected with BFF DNA (n=8/11 GHFF positive pools). The only two clade 2d.iii pools that were successfully sequenced (of four total) were included in the GHFF detections. With the individual level results, this further confirmed the strong virus-host association between clade 2d.iii and GHFF.”

And lines 752-775:

“To confirm the species contributing to under-roost pools, we screened RNA/DNA extracts using PCR targeting the cytochrome b mitochondrial gene (ref). Here, we chose to screen the entire column of extraction plates containing any CoV-positive samples that included all CoV positive wells (n=72) and neighbouring non-CoV positive wells (n=216). PCR was performed from the fecal extracts without cDNA synthesis using the Invitrogen Platinum SuperFi II PCR Master Mix (Thermo Fisher) with the primers L14724 (5'-CGAAGCTTGATATGAAAAACCATCGTTG-3') and H15149 (5'-AAACTGCAGCCCCTCAGAATGATATTTGTCCTCA-3') as per manufacturer's instructions. All amplicons of approximately the expected size (~425 bp) were sequenced using the Rapid Barcoding Kit 96 and R9.4.1 flowcells on a MinION Mk1C with rapid basecalling (Oxford Nanopore Technologies) targeting ~10,000 reads per samples. Following this, de-multiplexed sequence reads were filtered by length (between 250-450 bp) before mapping against a database of human and flying fox reference sequencing in Geneious Primer using Minimap2⁸⁰ with a minimum mapping rate of 25 reads. Consensus sequences were generated based on quality scores in Geneious Prime, which first excludes bases with a Phred quality score below 60% of the maximum possible score, then determines a majority consensus from the most frequent base among the remaining high-quality reads. Unmapped reads were de novo assembled in Geneious Primer and compared to NCBI GenBank using BLAST to determine any other species present. We then compared host data obtained through cytochrome b sequencing to the species recorded as roosting over the sheets in field data. Finally, host-species identification from the RNA sequencing data used for CoV WGS was also performed by mapping the filtered reads against a comprehensive database of cytochrome c oxidase I (COXI) gene sequences (available from <https://github.com/bachob5/MetaCOXI>).”

- If feasible, analyze individual fecal samples from the five pools that tested positive for Clade 2d.iii for more precise species identification.

Within the cytochrome b sequencing efforts described above, we screened each of the four pools (five was a previous data transcription error that has been corrected) that tested positive for clade 2d.iii. Only two produced a PCR product that was able to be sequenced, and both of these contained a mix of BFF and GHFF DNA (as outlined above). This is consistent with our findings from our individual bat results, where all clade 2d.iii positive individuals were GHFF. While analyzing the individual fecal samples within these pools would provide more precise species

identification, as previously highlighted, this was not feasible, however, our extensive effort to screen the existing under-roost pool extracts for host identification still yielded very meaningful and consistent results. That is, samples or pools shown to be solely from GHFF tested either positive for clade 2d.iii or negative across all clades—confirming the strong virus-host association with clade 2d.iii and GHFF. Furthermore, GHFF-positive pools that tested positive for other clades contained mixed-host DNA (i.e. both GHFF and BFF DNA), suggesting, alongside the individual results, the most parsimonious explanation is that the detections of the other clades came from the BHFF component of the sample.

Please see the comment above where we addressed these points in the manuscript.

Reviewer #3

Remarks to the Author: Peel et al. present a large-scale betacoronavirus surveillance study in Australian Pteropus bats. The paper is well written and provides a rich analysis of the collected data leading to some interesting conclusions. The authors clearly show a higher prevalence of the sampled betacoronaviruses in younger bats (that has been implicated in previous research) as well as highlight how different nobecovirus clades have different seasonal patterns, which is quite an important finding for the field.

Thank you for taking the time to review our manuscript.

I only have a few **minor suggestions which I outline below:**

1. The co-roosting patterns of these bats with other species is rather important when it comes to identifying the true host of the detected viruses (especially when it comes to pooled sampling). The authors refer to mitochondrial sequence data but do not present them in the paper. If host sequences are available for the pooled samples I would urge the authors to include these as supplementary in the current manuscript (for example perform some blast-based bat transcript identification analysis for their next-gen sequencing). If such sequencing data isn't available for the pooled samples the authors should expand on field observations about other potential species co-roosting with the Pteropus bats in the text.

We thank the reviewer for these helpful suggestions and agree on the importance of identifying the true host. Our research at these sites has yielded several complementary lines of evidence supporting the identification of Pteropus species as the primary hosts of these coronaviruses. First, we detected all but one of these viral clades in samples from individual flying foxes, confirming their presence in individuals from these species. Second, our under-roost sampling protocol was designed to minimize contributions from other species - we sampled immediately under the roost and collected samples within short time frames. Third, these coronavirus lineages are phylogenetically nested within bat nobecoviruses, with several matching previous nobecovirus detections from these Pteropus species.

Nonetheless, as described above, we have now undertaken extensive host sequencing and included these results in the main text and supplementary materials.

2. lines 154-155: I would urge against naming the new open reading frame ORFx. “ORFX” is a named protein in sarbecoviruses (<https://journals.asm.org/doi/10.1128/jvi.03079-15>) and using the same name can lead to confusion, especially since the virus groups are closely related.

Agreed. We have fixed the text and figures to simply refer to the possible ORF as ‘Unknown ORF’.

3. In table 2 I noticed that the location-specific hypothesis has the worst fit. Would that be explained by movement of individuals between the sampled locations? Do the authors have any evidence of that and how often would the expect movement between locations (and potential transmission of the viruses) to happen?

Yes, Pteropus individuals are capable of high mobility relative to the scale of our study area (>11,000 km²), and this likely contributes to the absence of meaningful spatial signal within the viral dynamics. Individual movements are highly variable, with some animals residing in the same roost for many months or even longer than a year, with others moving on within a few days. Indeed, a previous satellite estimated a daily turnover rate of 12% (Welbergen et al. 2020, ref #65 in manuscript).

To address these points, we have added the following point to the Discussion (Lines 488-493).

“While spatial sampling remains crucial for capturing potential clade differences across broad geographic regions and habitat types^{18,64} at the scale of our study area (>11,000 km²), we found that temporal surveillance rather than spatial surveillance was more informative for understanding coronavirus dynamics. This finding may be explained, in part, by the extreme mobility of Pteropus bats⁶⁵, and is therefore likely generalizable to other bat species known to travel long distances (such as annually migrating insectivorous bats)⁶⁶.”

4. Could the authors ensure that collection date, location (ideally map coordinates) and host species are available for all their submitted virus sequences? Either in the paper or (better) in their genbank entries.

We confirm that all submitted virus sequences include the collection date, latitude and longitude (to 2 decimal places), and host species to the finest resolution possible (i.e. to the host species level, where this information is definitive, or otherwise to the host genus where multiple species may have contributed). See Supp Table 3.

Remaining reviewer comments:

Reviewer #1: I would like to thank the authors for the huge efforts they have made to answer to the questions I had during the first round of review, even when my questions were more linked to scientific curiosity than a problem with the methodology they employed (for example with bat Ab decay). Especially I appreciate the demonstration of how the integrated approach can estimate more precisely the prevalence by pooling individual and pool positivity frequency. This is of major importance, and I wonder if this method should not be published separately to have a broader diffusion.

Reviewer #2: Thank you for addressing my comments thoroughly and attentively. I appreciate your thoughtful responses and am satisfied with them.

Reviewer #3: The authors have addressed all my concerns, I'm happy to recommend this paper for publication.

Our Response:

We thank the reviewers for their time and positive responses.

Regarding Reviewer 1's comments on whether the work showing how the integrated approach can estimate more precisely the prevalence by pooling individual pooled data could be published separately: We have already published the underlying methods (as referenced within the manuscript), and so feel that it is within scope to keep this within the current manuscript.